# TEtrimmer: a tool to automate the manual curation of transposable elements

Jiangzhao Qian [1,2], Hang Xue[3], Shujun Ou[4], Ludwig Mann[2], Jessica Storer[5], Lisa Fürtauer [6], Tony Heitkam[2], Mary C. Wildermuth [3], Stefan Kusch [1,7] ✉ & Ralph Panstruga [1] ✉

Transposable elements (TEs) are repetitive DNA sequences that move within genomes and play important roles in gene regulation and genome evolution. Accurate TE annotation in genomes is crucial for downstream analyses but challenging due to their sequence diversity and frequent fragmentation, including the occurrence of nested copies. We here present TEtrimmer, a tool that automates and replaces key steps of traditional manual curation of TEs. TEtrimmer combines phylogenetic tree analysis with the machine learning method DBSCAN to cluster TE sequences accurately and applies a sliding-window strategy to remove poorly conserved regions of TE-derived multiple sequence alignments. TEtrimmer also provides detailed report plots and features a graphical user interface (GUI) application. Tested on the genomes of six organisms belonging to various kingdoms of eukaryotic life and three simulated genomes, TEtrimmer consistently improved the identification of intact TEs compared to the established tools EDTA and RepeatModeler2.

Transposable elements (TEs) are selfish, repetitive DNA elements that can move within host genomes. TEs were first discovered by Barbara McClintock in maize in 1948[1]. Since then, TEs have been identified in almost all studied eukaryotic species where they occupy as either intact or fragmented copies a large proportion of many genomes, such as around 45% of the human genome[2], 53% of the zebrafish genome[3], and 85% of the maize genome[4].

TEs were long regarded as junk DNA. However, TEs play key roles in genome evolution, development, and immunity[5]. For instance, TEs can contain promoter and enhancer sequences to alter host gene expression[6]. In addition, TEs are a source of many regulatory genes, including long non-coding RNAs[7] and small RNAs[8]. Moreover, TEs can be drivers of evolutionary innovations. A recent example of TE-driven evolution is an *Alu* element insertion into an intron of the human *TBXT* gene, which appears to have contributed to tail-loss evolution in primates[9].

TEs are classified according to their transposition mechanism. Class I encompasses the retrotransposons, which copy themselves for genome insertion via an RNA intermediate. Class I retrotransposons mainly include three subclasses: LTR (long terminal repeat) retrotransposons, LINEs (long interspersed nuclear elements), and SINEs (short interspersed nuclear elements)[5,10]. An intact LTR retrotransposon always contains flanking terminal repeat sequences, while LINEs and SINEs are defined by their 3' poly(A), poly(T), or microsatellite tails[10]. Class II, on the other hand, comprises DNA transposons, which do not usually generate copies but rather excise themselves for insertion into another locus in the host genome. The most prominent DNA transposons include TIR (terminal inverted repeat) and Helitron elements[10]. Despite these common motifs, precise TE annotation remains challenging, mainly due to TE divergence and the existence of decayed, fragmented, nested, and low-copy TEs, especially in genomes with a high TE content[11,12].

[1]Unit of Plant Molecular Cell Biology, Institute for Biology I, RWTH Aachen University, Worringerweg 1, Aachen, Germany. [2]Chair of Molecular Botany, Institute for Biology I, RWTH Aachen University, Worringerweg 3, Aachen, Germany. [3]Department of Plant and Microbial Biology, University of California, Berkeley, CA, USA. [4]Department of Molecular Genetics, The Ohio State University, 592 Aronoff Laboratory, 318W 12th Avenue, Columbus, OH, USA. [5]Department of Molecular and Cell Biology, University of Connecticut, 67 North Eagleville Road, Unit 3179, Storrs, CT, USA. [6]Unit of Plant Molecular Systems Biology, Institute for Biology III, RWTH Aachen University, Worringerweg 1, Aachen, Germany. [7]Institute of Bio- und Geosciences - Bioinformatics (IBG-4), Forschungszentrum Jülich GmbH, Wilhelm-Johnen-Straße, Jülich, Germany. ✉e-mail: s.kusch@fz-juelich.de; panstruga@bio1.rwth-aachen.de

TE characterization involves two main steps, which are TE discovery and TE annotation[12]. Due to the repetitive nature of TEs, TE families can be represented by a TE consensus sequence library, also called TE library, which is constructed based on a multiple sequence alignment (MSA) of the TE copies from the respective reference genome[13]. The process of TE library construction is called TE discovery. The TE Hub Consortium[14] has collected 78 tools to date (June 2025) that are associated with TE library construction. The term TE annotation refers to the process of genome-wide TE identification to correctly define both intact and fragmented TE copies in the genome. Pre-constructed TE libraries are available for some species from repositories such as Repbase[15], Dfam[16], and RepetDB[17] to assist TE annotation. To date, Repbase contains the most comprehensive collection of manually curated high-quality TE libraries, but these have been behind a paywall since 2018.

TE annotation is typically based on three main strategies, i.e., repository-based, repeat-based, and structure-based methods. The most-used strategy is the repository-based approach[18], which takes advantage of public pre-constructed TE libraries for whole-genome TE annotation. Since closely related species share a common origin and exhibit genome sequence similarity, the established TE library can be used to identify homologous TEs in related genomes as well[19]. This strategy heavily relies on the availability and quality of a suitable TE library for the species of interest. However, manually curated high-quality TE consensus libraries are only available for a limited number of species after years of community efforts[16,20]. In addition, because the current versions of Repbase are only available via paid subscription, many researchers, particularly from smaller institutes and/or less wealthy countries, do not have access to suitable TE libraries. By contrast, the repeat-based strategy generates a custom TE library from the genome of interest[20]. This strategy mainly includes self-comparison, *k*-mer seeding[12], and short read-based clustering methods[21]. The self-comparison method identifies TEs by computationally intensive pairwise alignments between genomic sequences using algorithms such as RECON[22] and Grouper[23]. On the other hand, *k*-mer seeding methods like RED[24], RepeatScout[25], and P-Clouds[26] are generally faster and can efficiently handle large genomic datasets. However, the accurate clustering of TE sequences remains a challenge, and this method relies on high-quality genome assemblies. Poor genome quality can lead to inaccurate TE boundary definition, over-fragmentation of TE copies, and loss of TEs in incomplete assemblies[27]. Reference genome-free clustering of short reads, as implemented in RepeatExplorer2[21], will only cover the most abundant repeat families. Finally, the structure-based strategy can annotate TEs by recognizing specific structural hallmarks such as LTRs, TIRs, target site duplications (TSDs), and 3′ terminal genomic poly(A), poly(T), and microsatellite tails. A variety of algorithms, including LTR_FINDER[28], LTR_STRUC[29], LTRharvest[30], TIR-Finder[31], and SINE-Finder[32] facilitate structure-based TE annotation. The structure-based methods can efficiently annotate low-copy TEs and precisely define TE boundaries; however, high false-positive rates are associated with these approaches.

No single strategy or algorithm can comprehensively annotate TEs in a genome. Accordingly, various tools for de novo TE identification have been developed that integrate several annotation strategies, such as RepeatModeler2[33], EDTA[34], REPET[35], and others. However, even these tools do not approach the gold standard quality of manually curated TE libraries due to the frequent incompleteness and high rate of false-positive hits of the automatically generated TE library[36,37]. Accordingly, manual curation of TEs is still mandatory for a detailed analysis of TE diversity and evolution within a genome[13,37].

Manual curation of TEs can be divided into five major steps: Searching similar sequences in the genome, e.g., via BLAST (Basic Local Alignment Search Tool), extending BLAST hit ends, extracting TE sequences, aligning sequences, clustering the TE-derived MSA, and trimming the MSA[13,38]. In brief, each of the TE consensus sequences from a de novo discovery pipeline is used for a nucleotide BLAST (BLASTN) query against the reference genome, and the coordinates of BLASTN hits are arbitrarily extended at both ends to include the potential TE boundaries. Then, the TE sequences are extracted based on the extended coordinates from the corresponding genome and subsequently aligned to generate MSAs. Due to the high sequence similarity among closely related TEs, a single MSA may exhibit multiple distinct alignment patterns. To separate these TE variants, the TE sequences within each MSA are manually clustered and separated based on their sequence relatedness. Finally, MSAs are trimmed by removing regions of low conservation[13,37,38]. The refined MSA is then used to generate a new TE consensus sequence.

The manual curation of TEs can yield a gold standard custom TE library for the genome of interest, but is very time-consuming to establish, and requires TE specialists with an in-depth understanding of TE structure and biology[39]. Accordingly, the comparability and reproducibility of TE libraries from manual curation may be limited by the level of expertise of the curators[40].

Some software tools, like EarlGrey[40] and MCHelper[39], have been developed to partially automate and assist manual curation of TEs. EarlGrey mainly combines RepeatModeler and TEstrainer (https://github.com/jamesdgalbraith/TEstrainer) for TE library construction. TEstrainer is the core module to automate most manual curation steps, but not the clustering of MSAs. TEstrainer attempts to bypass the MSA clustering by only including the top 20 BLASTN hits, which is insufficient for highly diverged TEs. MCHelper includes an MSA clustering method but lacks effective MSA cleaning ability. In essence, both EarlGrey and MCHelper can improve the precision of TE annotation in any given organism, but do not yet approach the quality of manually curated TEs. Here, we introduce a user-friendly tool called TEtrimmer that automates the manual curation of TEs. TEtrimmer can efficiently cluster MSAs, clean MSAs, and precisely define TE boundaries. TEtrimmer provides comprehensive report plots and a graphical user interface (GUI) application to support the rapid inspection and improvement of results, which closes the gap to manual curation.

## Results

### Overview of TEtrimmer

Manual curation is crucial for high-quality TE annotation. However, curation can involve hundreds to thousands of TE consensus sequences for each genome and is therefore time-consuming and requires expert TE knowledge. TEtrimmer is a software designed to automate the manual curation of TEs. The input of TEtrimmer can be the output TE consensus libraries from de novo TE discovery tools like EDTA, RepeatModeler2, and REPET, and/or TE libraries from closely related species. To allow multi-thread analysis, TEtrimmer saves each set of input sequences into separate files, each representing a single sequence (Fig. 1). For each separated sequence, TEtrimmer automatically performs BLASTN searches against the respective genome[41]. It uses the retrieved BLAST hits for MSA generation, MSA clustering (Fig. 2), MSA sequence extension, and MSA cleaning (Fig. 3).

Thereafter, TEtrimmer employs the clustered, extended, and cleaned MSAs to generate consensus sequences for the definition of putative TE boundaries. Then, potential terminal repeats are identified, and a prediction of open reading frames (ORFs) and protein domains is conducted based on the protein families (PFAM) database[42]. Subsequently, TE sequences are classified, and an output evaluation is performed mainly based on the existence of terminal repeats and the full-length BLASTN hit numbers (Supplementary Table 1). Consequently, two TE consensus library files are produced by TEtrimmer. The first file contains all output sequences, while the second file is generated by removing duplicated sequences via CD-HIT-EST[43] (Fig. 1; see "Methods" for further details).

TEtrimmer provides a series of report plots for each output sequence to enable detailed inspection and to illustrate the differences

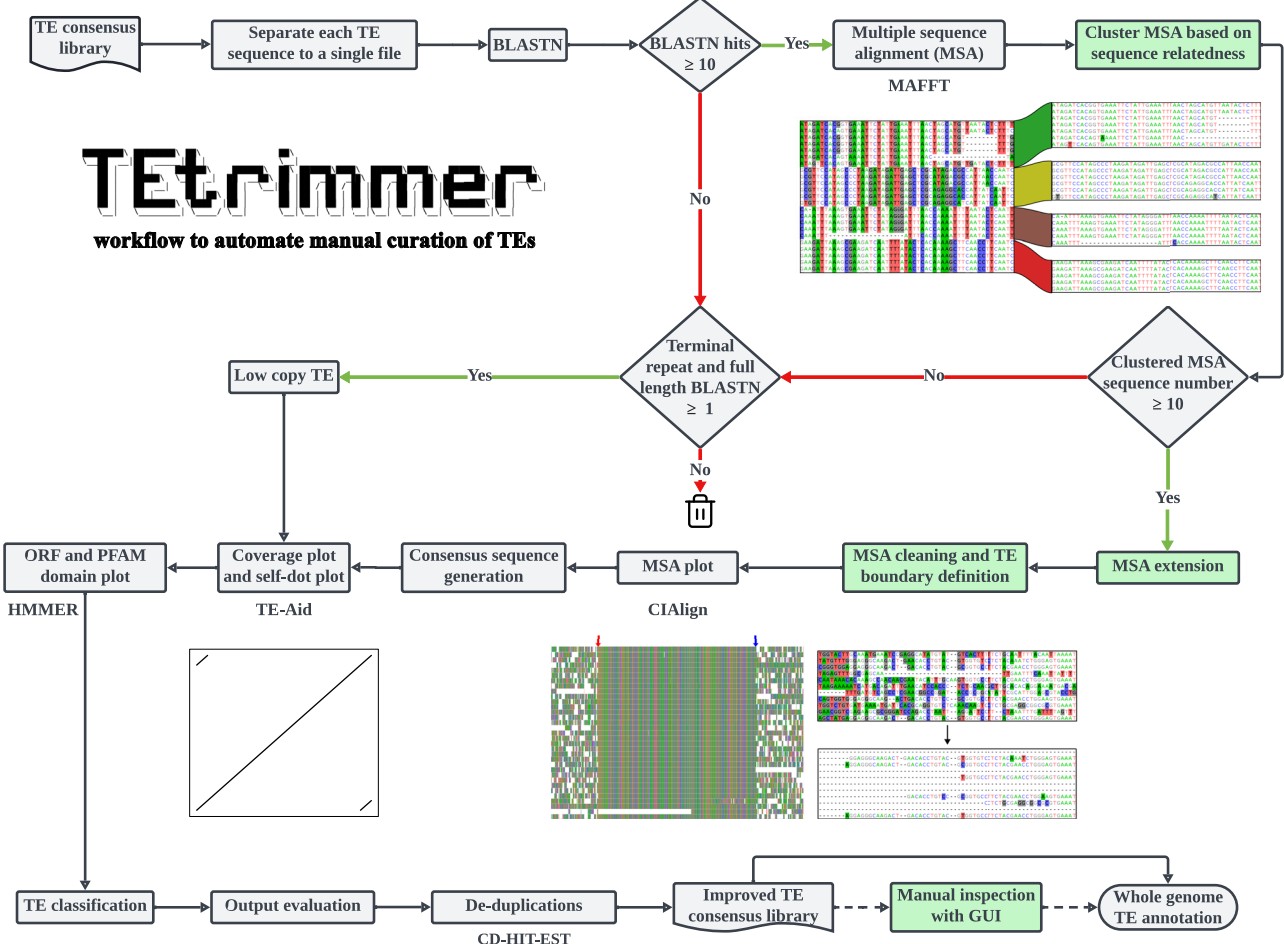

**Fig. 1 | TEtrimmer workflow.** TEtrimmer is a multi-thread tool to automate the manual curation of TEs. It performs BLASTN searches of TE sequences retrieved from any TE consensus library against the respective genome and facilitates the generation of MSAs, MSA clustering, MSA sequence extension, MSA cleaning, TE boundary definition, and TE classification. Moreover, TEtrimmer provides detailed report plots and offers a graphical user interface (GUI) application to enable inspection and improvement of each output. Key features of TEtrimmer are highlighted by green background color. The trash bin icon was downloaded from Feather (https://feathericons.com/) and is licensed under MIT license (https://github.com/feathericons/feather?tab=MIT-1-ov-file#readme).

in the TE consensus sequences before and after TEtrimmer analysis (Fig. 4). Users can choose to apply the TEtrimmer-generated TE consensus library directly for genome-wide TE annotation or take advantage of the supplied GUI application (Fig. 5) to inspect and curate the TE library, which can generate a gold standard manual curation-level TE library. We highly recommend that users always revise sequences not classified as Perfect or Good for optimal TE library quality. A detailed description of each step can be found in the Methods section, and detailed tutorial videos are available at (https://tehub.org/en/Tutorials).

### TEtrimmer efficiently clusters sequences in the MSA
The TEtrimmer tool initially performs a BLASTN search for each input TE consensus sequence against the corresponding genome (Fig. 1). If more than ten hits are obtained, an MSA is established; otherwise, the sequence is considered a potential low-copy TE. Due to the often high-sequence similarity among closely related TE variants, this initial BLASTN search typically retrieves multiple variants for each query TE. These can be visualized by distinct alignment patterns of the TE sequences corresponding to the BLASTN hits (Fig. 2A). A labor-intensive task of the traditional manual curation of TEs is to separate these TE variants within the MSA according to sequence relatedness. TEtrimmer combines a phylogenetic tree approach (Fig. 2B) and a machine learning method (Density-Based Spatial Clustering of

Applications with Noise, DBSCAN)[44] to separate different alignment patterns automatically.

To illustrate the TEtrimmer clustering ability, one LTR retro-transposon from *Drosophila melanogaster* identified by RepeatModeler2 was selected as an example. The BLASTN search of the selected TE sequence against the *D. melanogaster* genome yielded 99 hits. After the extraction of the BLASTN hit sequences and generation of an MSA, 101 divergent columns were selected in the original MSA and combined into a new MSA of pseudo-sequences representing the deviating regions in the original MSA (Fig. 2A, left part). Divergent columns are defined as columns where the proportion of the predominant nucleotide in that column is below 0.8. The combined MSA based on the divergent columns was used to construct an un-rooted maximum-likelihood tree (Fig. 2B), which was employed to calculate a relative tree branch distance matrix (Source Data file). Finally, DBSCAN used the distance matrix for MSA clustering, which yielded fourteen sub-MSAs (clusters), likely corresponding to different TE variants in the original MSA (Fig. 2A, right part and Fig. 2B). Thirteen of these clusters, clusters 1 to 13, had clearly grouped sequences. The fourteenth cluster was dubbed Noise, which means the sequences in this cluster could not be sub-clustered further. The TEtrimmer MSA clustering efficiency was additionally supported by a principal component analysis (PCA) plot based on the phylogenetic tree relative branch distance matrix (Fig. 2C). In the case of clusters 1 to 13, the sequences in each cluster grouped adjacent to each other but

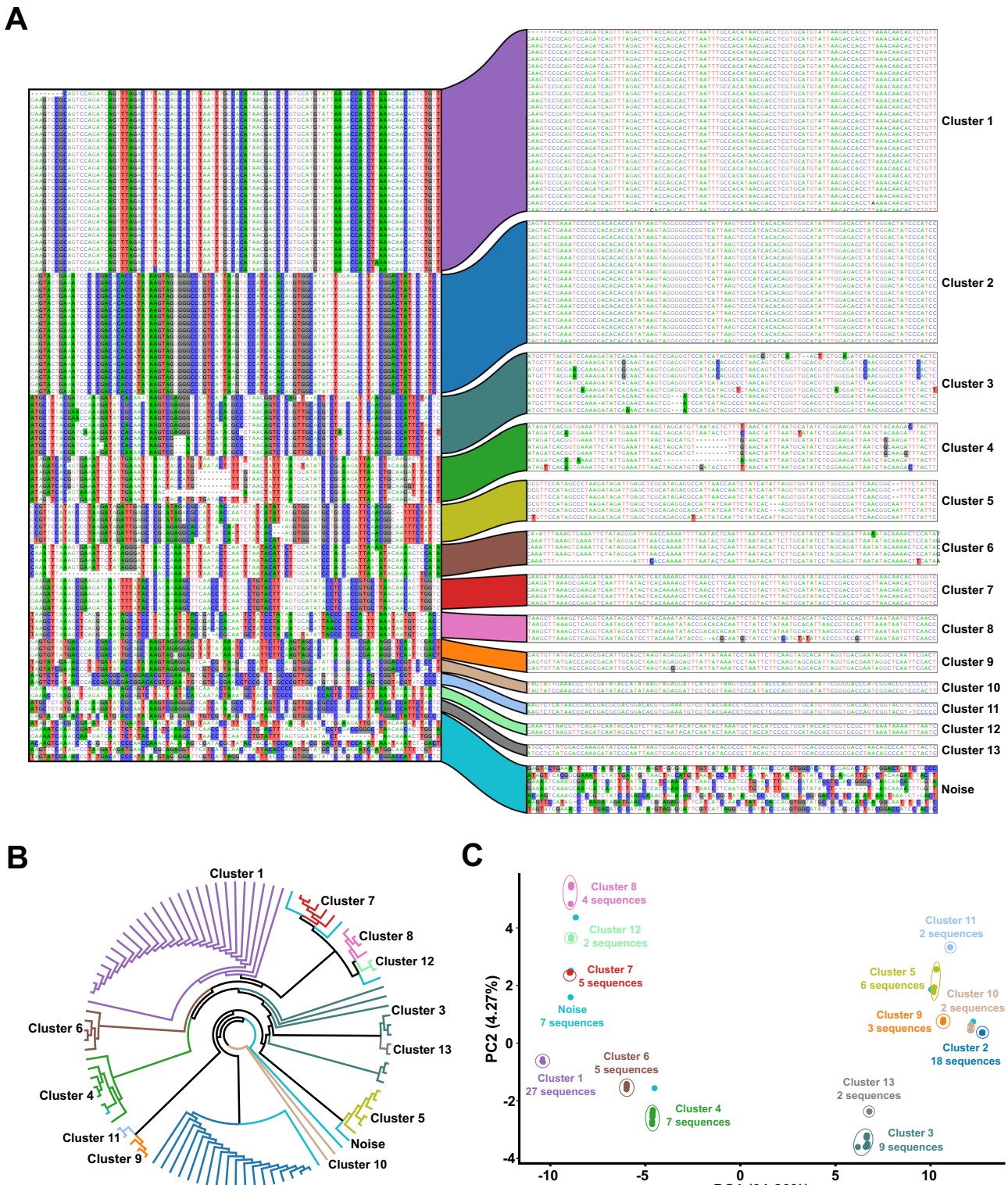

**Fig. 2 | TEtrimmer efficiently clusters sequences within the MSA.** The *D. melanogaster* LTR retrotransposon (named rnd-1-family-39), identified by RepeatModeler2, was chosen as an example to illustrate the clustering routine. TEtrimmer performed a BLASTN search of this LTR retrotransposon against the *D. melanogaster* genome, established an MSA with the respective BLASTN hit sequences, and conducted MSA clustering. **A** MSA of 99 sequences. The left part represents the divergent columns in the MSA before clustering. The right area displays the MSAs after clustering by TEtrimmer based on the relative branch distance-based DBSCAN analysis (epsilon = 0.1, min sample number = 2) of the phylogenetic tree shown in (**B**). Nucleotides are colored with: A, green; C, blue; G, black; T, red. Nucleotide background colors in the MSA represent sites where the proportion of the respective nucleotide in that column is below 0.4. The cluster names given here were consistently used in (**B**) and (**C**). Each color represents the corresponding cluster. **B** The un-rooted maximum likelihood phylogenetic tree generated by IQ-TREE is based on the MSA before clustering. Cluster labels and colors are consistent with (**A**). **C** PCA plot calculated using the phylogenetic tree relative branch distance matrix (**B**). The percentage of variance explained by the respective principal components PC1 and PC2 is indicated on the *x*- and *y*-axis, respectively. Cluster labels and colors correspond with (**A**). The number of sequences within each cluster is indicated. Raw data: Source Data file.

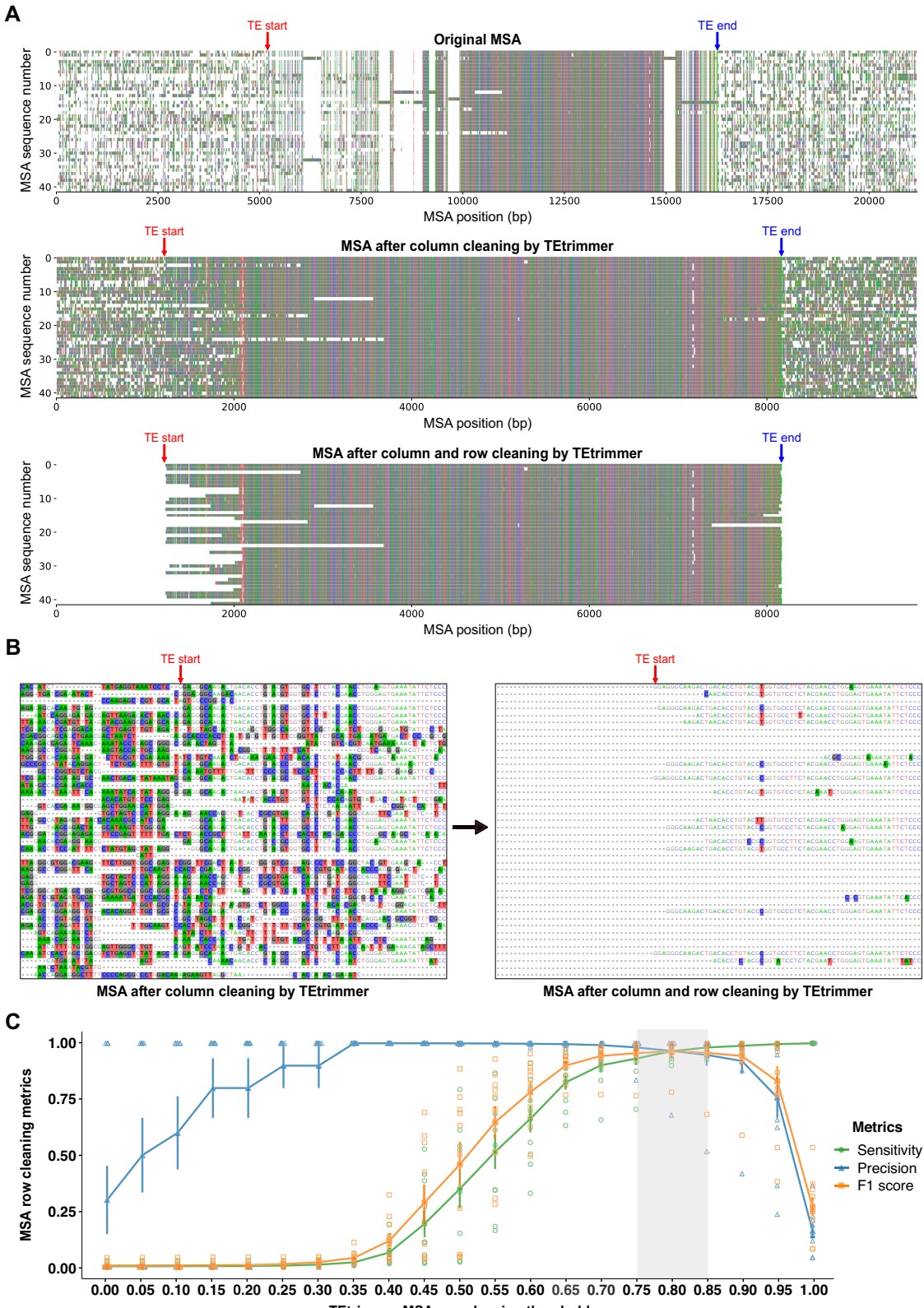

were separated from the sequences of the other clusters. Regarding the Noise cluster, its seven sequences were randomly distributed across the entire PCA plot, indicating their low relatedness. This rather intricate example of a complex MSA pattern illustrates that TEtrimmer can efficiently cluster TE sequences in an MSA.

Compared to young, recently generated TE copies, highly diverged TE copies are more difficult to cluster. TEtrimmer also

addresses this particular scenario, as illustrated here with the DNA transposon rnd-1-family-420 that was identified by RepeatModeler2 in the powdery mildew fungus *Blumeria hordei* (Supplementary Fig. 1). For the DNA transposon rnd-1-family-420, 97 out of 100 TE sequences from the corresponding MSA were grouped into the Noise cluster (Supplementary Fig. 1D), since the sequence number in the Noise cluster was larger than 15 and occupied over 60% of the total

**Fig. 3 | TEtrimmer succeeds in removing lowly conserved regions from MSAs.** The *B. hordei* LINE element rnd-1-family-34, identified by RepeatModeler2, was chosen. **A** The boundaries of the selected LINE sequence identified after BLASTN search against the *B. hordei* genome, sequence extension, and MSA generation are indicated by TE start (red) and TE end (blue). Nucleotides are represented with colored bars (A, green; C, blue; G, black; T, red); gaps are indicated as blank regions. Top panel: original MSA before cleaning, containing many gappy columns and noisy rows. After MSA column cleaning by the TEtrimmer function `remove_gap_column`, the majority of the gappy columns are removed (middle panel). Then, TEtrimmer cleans sequences in the MSA row by row using the TEtrimmer function `crop_end_by_divergence` (bottom panel), removing lowly conserved regions (bottom panel). **B** The left and right panels are both the magnified MSA regions near the selected LINE element left boundary, indicated by TE start (red). The left panel illustrates the MSA before, and the right panel after row cleaning. Nucleotide background colors in the MSA represent sites where the proportion of the respective nucleotide is below 0.4. **C** Ten TEs were randomly selected from the RepeatModeler2 consensus libraries of *B. hordei*, *D. melanogaster*, *D. rerio*, and *O. sativa*. After BLASTN searches of the selected sequence against the corresponding genomes, sequence extension, and MSA column cleaning, the generated MSAs were used for benchmarking. Manual cleaning was performed to serve as a reference to enable assessment of the TEtrimmer MSA cleaning performance. The MSAs were cleaned by the TEtrimmer cleaning function `crop_end_by_divergence` using cleaning thresholds ranging from 0 to 1 and a sliding window size of 40 bp. A confusion matrix analysis was conducted to evaluate the TEtrimmer cleaning performance. The *x*-axis shows the cleaning threshold used by the TEtrimmer function `crop_end_by_divergence`; the *y*-axis displays the confusion matrix score for the metrics sensitivity (green), precision (blue), and F1 score (orange). Standard error bars were calculated based on *N* = 10 TE sequences and data points depict the respective arithmetic means. The grey-shaded box indicates the threshold range where all three metrics scores are above 0.93. Raw data: Source data file.

sequences of the original MSA (see "Methods"). This Noise cluster was used for all subsequent analyses, such as MSA extension and cleaning. Finally, TEtrimmer generated a longer and more accurate TE consensus sequence (Supplementary Fig. 1A, B, C, and E). In general, if the TE sequences are too diverged, the TEtrimmer cluster module will classify most of the sequences from the MSA as noise and place these sequences into a Noise cluster (see "Methods" for further details on the MSA clustering).

## TEtrimmer efficiently extends and precisely cleans clustered MSA sequences

As the TE-related sequences retrieved by BLASTN searches are often incomplete, the sequences in the clustered MSAs typically need to be extended. TEtrimmer offers a convenient function for the expansion of sequence ends in the MSAs, which is iteratively used by the tool until the sequences cover the putative TE boundaries (Fig. 1; see "Methods" for further details on the sequence extension procedure). However, sequence extension beyond the actual TE borders usually results in non-TE sequences added to the MSA. These non-TE sequences are typically dissimilar and thus produce badly aligned noisy regions in the MSA (Fig. 3A). Cleaning these regions is a time-consuming step of the manual curation process. The cleaning (trimming) module of TEtrimmer efficiently addresses this task by removing gappy columns and the majority of lowly conserved regions within rows of MSAs after sequence extension. We demonstrate this capability by a representative MSA based on extended sequences derived from a *B. hordei* LINE element, rnd-1-family-34 (Fig. 3A). After MSA column cleaning by the TEtrimmer function `remove_gap_columns`, the majority of gappy columns within the original MSA were eliminated (top and middle panel of Fig. 3A). Subsequently, the remaining noisy regions of the MSA were cleaned row by row by the TEtrimmer function `crop_end_by_divergence` (middle and bottom panels of Fig. 3A and B). Following column and row cleaning of the MSA, TE boundaries (TE start and TE end) are determined by TEtrimmer (Fig. 3A and B, see "Methods" for details). To evaluate the MSA cleaning performance, we compared manual cleaning to TEtrimmer-based cleaning for the above-mentioned LINE element. The effectiveness of cleaning conducted by TEtrimmer was assessed quantitatively using a customized confusion matrix[45] (Supplementary Table 2), which showed a cleaning sensitivity of 0.977 and a precision of 0.999. This means that TEtrimmer correctly identified and removed 97.7% of the manually cleaned MSA regions, and that 99.9% of the regions it removed were confirmed by manual cleaning. An additional MSA row cleaning function of TEtrimmer, not illustrated here, is `crop_end_by_gap`. In the TEtrimmer analysis process, this is an additional option preferentially used for LINE elements to deal in particular with their frequently truncated 5′ regions. In contrast to the `crop_end_by_divergence` function, which is based on nucleotide proportions, `crop_end_by_gap` takes advantage of gap information for the cleaning process (see "Methods"

for further details). For instance, combining both `crop_end_by_divergence` and `crop_end_by_gap` functions, TEtrimmer effectively cleaned the MSA of a highly diverged LINE element from *Danio rerio* (rnd-5-family-5398), which had only a few (7) intact copies (Supplementary Fig. 2).

To test the cleaning performance across various types of TEs and different species, we randomly selected ten TEs from *B. hordei*, *D. melanogaster*, *D. rerio*, and *Oryza sativa*, including LTR retrotransposons, LINEs, and DNA transposons. After conducting BLASTN searches of the selected TE sequences against the corresponding genomes, sequence extension, MSA generation, and removal of gappy columns, we performed both manual and TEtrimmer-based MSA cleaning, the latter using the function `crop_end_by_divergence` with different thresholds. A detailed explanation of the cleaning thresholds can be found in the Methods section. TEtrimmer showed optimal row cleaning performance with a set threshold between 0.75 and 0.85, yielding averaged sensitivity and precision scores above 0.959 and 0.971, respectively (Fig. 3C and Source Data file). The results highlight the excellent MSA cleaning ability of TEtrimmer across various TE types from different species. Accordingly, the software name TEtrimmer relates to the advanced MSA trimming ability of the tool.

## TEtrimmer provides detailed report plots and summary tables for each discovered TE

Detailed report plots and summary tables for each output facilitate the assessment of outputs and highlight differences in TE consensus sequences before and after TEtrimmer analysis (Fig. 4 and Supplementary Table 3). A *B. hordei* LTR retrotransposon, named ltr-1-family-22 and initially identified by RepeatModeler2, was used for the illustration of the report plots (Fig. 4 and Source Data file). The first plot presents the MSA columns around TEtrimmer-defined TE boundaries. The precise boundary positions are denoted by colored arrows (Fig. 4A). This plot can help to inspect the existence of TSDs and the accuracy of TE boundary definition. In the example, the marked boundaries clearly separate the highly divergent from the conserved regions, and TSD sequences can be identified (e.g., GCACG for the first sequence in Fig. 4A), which represents a correct boundary definition as judged by manual inspection. While the TE boundary plot only shows the ends of the MSA, the second plot displays the entire MSA (Fig. 4B), which enables users to verify the alignment quality of the full MSA after TEtrimmer analysis. The third type of plot illustrates the divergence of BLASTN hits against the TE consensus sequence and their coordinates along the TE consensus sequence (Fig. 4C). For instance, the selected exemplary LTR retrotransposon initially showed only one full-length BLASTN hit (Fig. 4C left panel), but 77 full-length BLASTN hits were recovered after TEtrimmer analysis, which indicates a substantial improvement (Fig. 4C right panel). The fourth type of plot is a self-dot plot, which can reveal TE sequence features such as flanking LTRs and TIRs (Fig. 4D). The example self-dot plot after TEtrimmer analysis

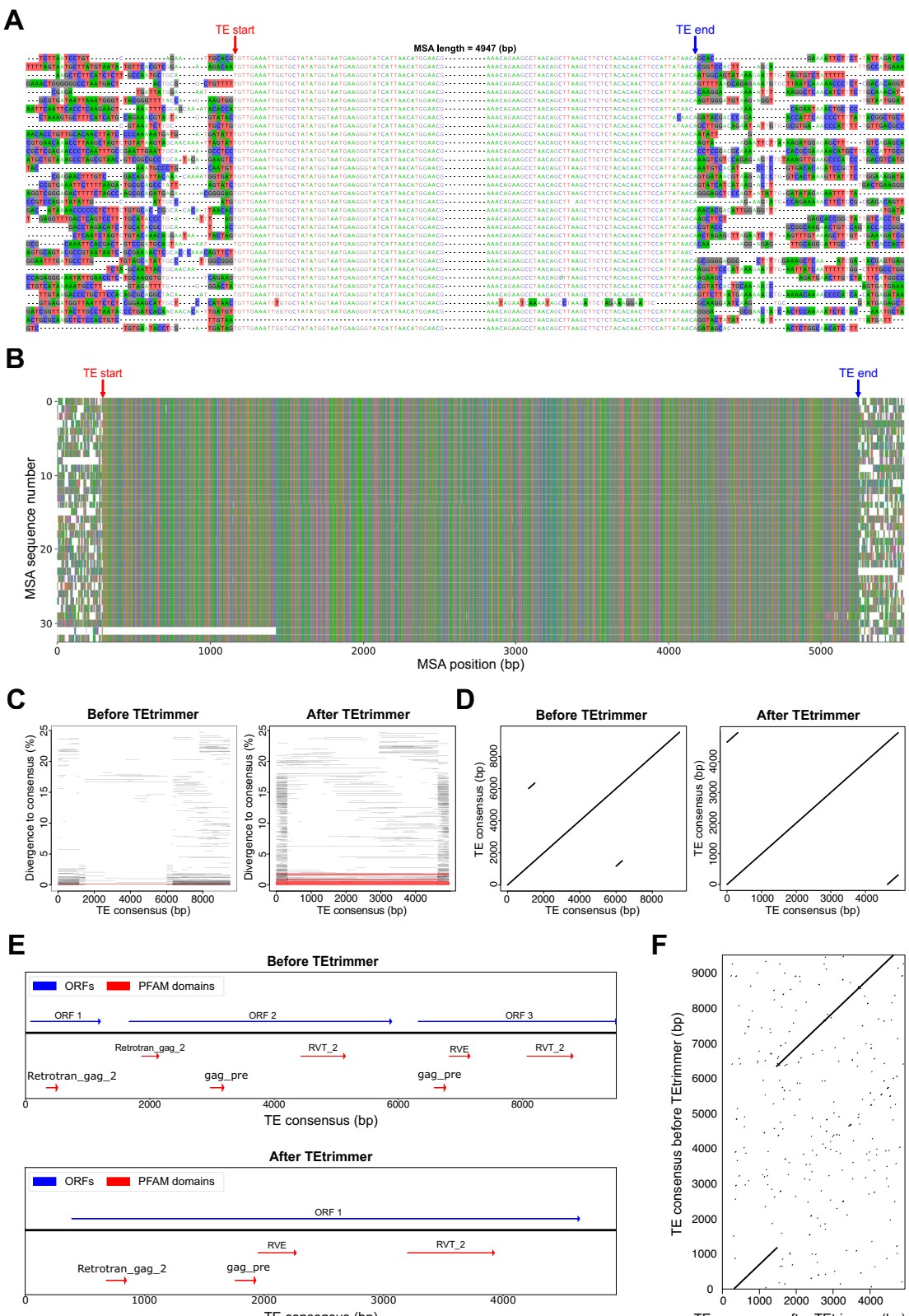

shows flanking LTRs from bp 1 to 310 and bp 4638 to 4947 of the output consensus sequence (Fig. 4D right panel), but only internal repeat regions (from bp 1167 to 1506 and bp 6003 to 6342), indicative of a faulty TE, were identified for the input consensus sequence (Fig. 4D left panel). Besides BLASTN hits and sequence features, ORF and protein domains are also crucial for evaluating a TE consensus sequence. TEtrimmer provides a fifth type of plot to display the ORF

and PFAM domains of the consensus sequence before and after TEtrimmer analysis (Fig. 4E). For instance, after TEtrimmer analysis, ORF 2 was eliminated by deleting the sequence region from bp 1167 to 6002 of the input consensus sequence. Additionally, ORF 1 and ORF 3 of the input consensus sequence (Fig. 4E, upper panel) were fused into the single ORF 1 of the output consensus sequence (Fig. 4E, lower panel). The final plot is a dot plot comparing TE consensus sequence

**Fig. 4 | TEtrimmer provides comprehensive report plots for each discovered TE.** The *B. hordei* LTR retrotransposon named ltr-1-family-22, initially annotated by RepeatModeler2 and re-analyzed with TEtrimmer, is shown as an example (consensus sequences: Source data file). **A** The alignment of 33 sequences in total shows the TE boundary regions of the MSA (100 nucleotide sites are displayed for each side) after TEtrimmer analysis. Nucleotide background colors (A, green; C, blue; G, black; T, red) in the MSA represent sites where the proportion of the respective nucleotide is below 0.4. The TE boundaries defined by TEtrimmer are indicated as TE start (red) and TE end (blue). The two boundary regions were artificially connected by ten gaps (-), and the total length of the MSA is indicated on the top. **B** The plot shows the entire MSA after TEtrimmer analysis. The TE boundaries are indicated as in (**A**). Nucleotides are represented with colored bars; gaps are indicated as blank regions in the plot. The *x*-axis gives the nucleotide position (in bp) within the MSA. **C** The panels show a BLASTN plot before (left

panel) and after (right panel) TEtrimmer analysis. The *x*-axis indicates the TE consensus nucleotide position (in bp), and the *y*-axis is the sequence divergence in percent compared to the TE consensus sequence. Each line indicates a BLASTN hit; red lines highlight hits with a sequence divergence below 1.5% and a sequence coverage >90%. **D** Self-dot plots before (left panel) and after (right panel) TEtrimmer analysis. The axes show the nucleotide position (in bp) in the TE consensus sequence. **E** The plot shows a genomic map of open reading frames (ORFs; blue arrows) and PFAM domains (red arrows) predicted in the TE consensus sequence before (upper panel) and after (lower panel) TEtrimmer analysis. **F** Dot plots (window size = 25 bp, threshold = 50; the threshold represents the minimal sum of substitution scores in a defined window required for a dot to be plotted) of the TE consensus sequence before (*y*-axis) and after (*x*-axis) TEtrimmer analysis. Axes show the nucleotide position (in bp).

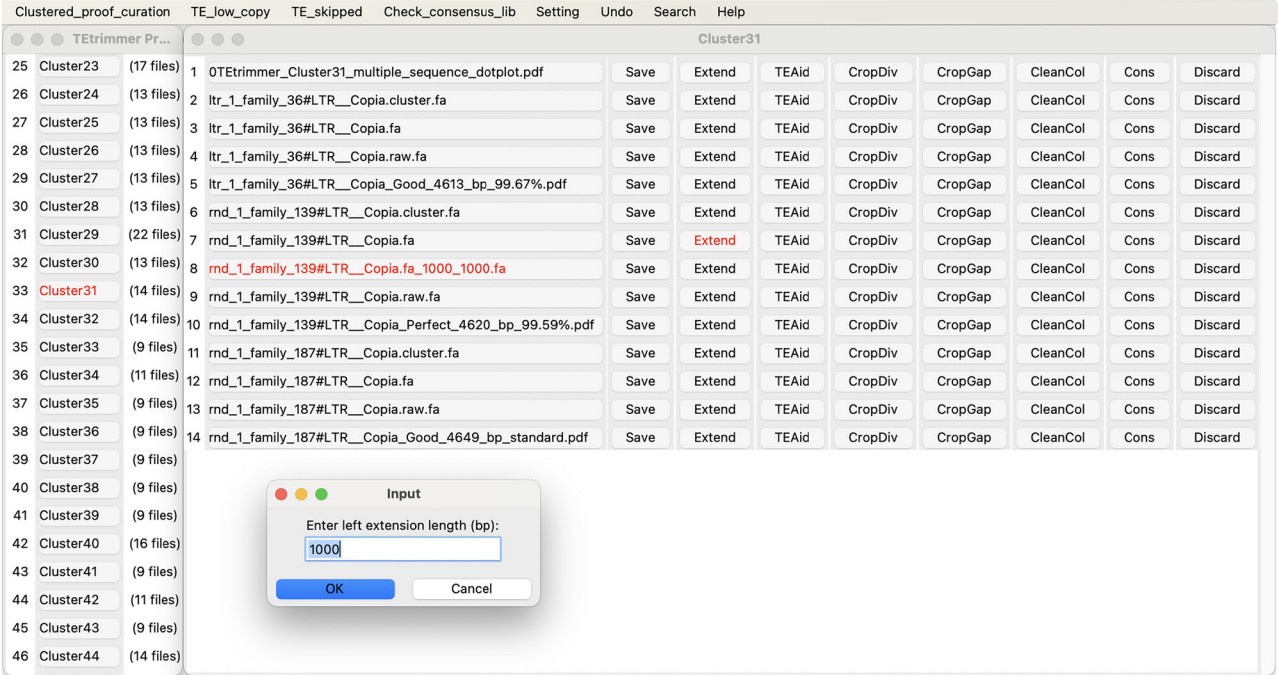

**Fig. 5 | TEtrimmer provides a graphical user interface (GUI) to inspect and improve discovered TEs.** Four files are associated with each consensus sequence (see main text for further details). All files are grouped when their corresponding consensus sequences have >90% identity (Cluster). Users can inspect each result by selecting the corresponding file name. TE sequences in the MSA can be extended by

activating the Extend button. The TEAid button helps to generate the interactive report plot (Supplementary Fig. 4). The buttons CropDiv, CropGap, and CleanCol relate to the MSA cleaning functions `crop_end_by_divergence`, `crop_end_by_gap`, and `remove_gap_columns`, respectively. The button Cons can be used to generate a consensus sequence based on the corresponding MSA.

similarity before and after TEtrimmer analysis (Fig. 4F). The dot plot highlights both similar and different regions between the two sequences. For example, the regions from bp 0 to 1166 and bp 6343 to 9508 of the input sequence were identical with bp 311 to 1477 and bp 1478 to 4637, respectively, of the output sequence. In addition to the mentioned plots, TEtrimmer provides a summary table compiling the analysis results, such as the numbers of BLASTN hits, sequence lengths, TE classifications, and the existence of terminal repeats (Supplementary Table 3). The summary table helps to browse and index analysis results conveniently. By combining the report plots and the summary table, users can quickly evaluate the accuracy of TEtrimmer outputs and their differences from the input TE consensus sequence.

## TEtrimmer provides a graphical user interface for inspecting and improving discovered TEs

To examine and improve the de novo-generated TE consensus libraries established by TEtrimmer further, a GUI application is provided, which

allows inspecting and correcting outputs conveniently. Four files are associated with each consensus sequence in the GUI: (1) the original MSA based on the BLASTN search of the input TE consensus sequence against the corresponding genome, (2) the MSA after clustering, sequence extension, and MSA cleaning but before TE boundary definition, (3) the MSA with defined TE boundaries used to generate the final TE consensus sequence, and (4) several report plots (Fig. 4). To group sequences with high similarity, TEtrimmer calculates the nucleotide identity among all TE consensus output sequences. All four files mentioned above that relate to highly similar TE consensus sequences (identity >90%) are placed together into one TE cluster and listed by the GUI application (left window in Fig. 5).

For instance, after TEtrimmer analysis, the TE sequences ltr_1_family_36, rnd_1_family_139, and rnd_1_family_187, which share more than 99.59% identity, and their associated files were placed into one TE cluster (Cluster 31; right window in Fig. 5). Moreover, multiple sequence dot plot files were provided to help visualizing consensus sequence similarities within a TE cluster (Supplementary Fig. 3). The

**Table 1 | Genomes and TE reference libraries used to benchmark TEtrimmer**

| Organism | TE reference library | Genome version | Genome link |
|---|---|---|---|
| *Blumeria hordei* | In-house manually curated | BGH_DH14_v4 | https://www.ncbi.nlm.nih.gov/datasets/genome/GCA_900239735.1/ |
| *Drosophila melanogaster* | Repbase | Release 6 plus ISO1 MT | https://www.ncbi.nlm.nih.gov/datasets/taxonomy/7227/ |
| *Danio rerio* | Repbase | GRCz11 | https://www.ncbi.nlm.nih.gov/datasets/genome/GCF_000002035.6/ |
| *Oryza sativa* | Repbase | version_7.0 | http://rice.uga.edu/pub/data/Eukaryotic_Projects/o_sativa/annotation_dbs/pseudomolecules/version_7.0/all.dir/ |
| *Zea mays* | Repbase | Zm-B73-REFERENCE-NAM-5.0 | https://www.ncbi.nlm.nih.gov/datasets/genome/GCF_902167145.1/ |
| *Homo sapiens* | Dfam | GRCh38.p14 | https://www.ncbi.nlm.nih.gov/datasets/genome/GCF_000001405.40/ |
| Simulated genome 1 | Simulated | 50 Mb with 50% TEs | https://github.com/qjiangzhao/TEtrimmerPaperFile/blob/main/simulated_genomes_data |
| Simulated genome 2 | Simulated | 100 Mb with 50% TEs | https://github.com/qjiangzhao/TEtrimmerPaperFile/blob/main/simulated_genomes_data |
| Simulated genome 3 | Simulated | 100 Mb with 75% TEs | https://github.com/qjiangzhao/TEtrimmerPaperFile/blob/main/simulated_genomes_data |

multiple sequence dot plots help users to inspect and compare similar outputs conveniently. Clustered files can be accessed by selecting the corresponding file names. Users can quickly assess each consensus sequence via the corresponding report plots file (Fig. 4) (e.g., rnd_1_family_139#LTR_Copia_Perfect_4620_bp_99.59%.pdf). If the TEtrimmer output appears to be inaccurate, users can easily modify and improve it using the provided MSA files and functions. For example, if the sequence extension is insufficient in the view of the user, it can be manually elongated by activating the Extend button (right window in Fig. 5). The default extension size is 1000 bp for both ends, and these values can be modified separately for each side. After extension, a new file will appear with the extended sequences (e.g., rnd_1_family_139#LTR_Copia.fa_1000_1000.fa, with 1000 bp extension on each side). Furthermore, to assist in identifying TE boundaries after manual extension, we integrated plot functions into the GUI application, allowing users to generate interactive report plots (Supplementary Fig. 4) by activating the TEAid button. This button creates plots similar to those generated by the tool TE-Aid[13], and the corresponding function was rewritten in Python for this application, based on TE-Aid. Additionally, the buttons CropDiv, CropGap, and CleanCol can be used to clean the MSA efficiently, which represent the same functions used for automated MSA cleaning in the TEtrimmer pipeline (Fig. 3). In conclusion, the TEtrimmer GUI application integrates various files and functions associated with sequence extension, MSA cleaning, boundary identification, and visualization of TE consensus sequences, enabling users to examine and improve the TE consensus library conveniently. We highly recommend that users revise TEs not classified as Perfect and Good to ensure a high quality of their final TE libraries. With the TEtrimmer GUI, the manual curation level of the TE consensus library can be readily achieved.

## TEtrimmer improves the recovery of intact TEs compared to EDTA and RepeatModeler2

We tested the TEtrimmer performance on six organisms belonging to various kingdoms of eukaryotic life and three simulated genomes, at a length of 50-100 Mb and a TE content of 50-75%. The six organisms were *B. hordei* (barley powdery mildew fungus), *D. melanogaster* (fruit fly), *D. rerio* (zebrafish), *O. sativa* (rice), *Zea mays* (maize), and *Homo sapiens* (human). (Table 1). The simulated genomes were created according to previously described methods[46], with random subsampling of BeetRepeatDB[47] as the reference TE library (see "Methods" for details). Benchmarking was initially conducted by directly comparing reference TE consensus libraries with de novo-generated libraries derived from EDTA, RepeatModeler2, and both these tools after additional TEtrimmer analysis (consensus library after de-duplication). For this approach, we adopted the benchmarking method

reported by Flynn and co-workers[33]. Four benchmarking levels were assigned to evaluate sequence matches between the reference libraries and the de novo-generated libraries, including Perfect, Good, Present, and Not found. For the analysis, LTR retrotransposons were separated into flanking LTRs and LTR retrotransposon internal sequences (designated as LTR-internal) for all de novo-generated and reference libraries. We found that up to 74% of the Perfect families were comprised of flanking LTRs (Source Data file). To mitigate the potential bias caused by these sequences, we eliminated all flanking LTRs for this benchmarking (see "Methods" for details).

The comparison between TEtrimmer results based on EDTA (EDTA+TEtrimmer) and the EDTA consensus library (EDTA lib) revealed that TEtrimmer improved the identification of Perfect transposable element (TE) families by recovering 1.3- to 29.8-times more Perfect TE families across all six tested organisms and the three simulated genomes (Fig. 6). Specifically, EDTA+TEtrimmer consistently generated more DNA and LINE elements in the Perfect family category but fewer LTR retrotransposons in the case of *O. sativa* (Fig. 7). EDTA struct+lib was generated by combining the EDTA consensus library with all TE sequences discovered by the EDTA structure-based module, followed by the removal of nearly identical sequences. EDTA+TEtrimmer revealed more Perfect families than EDTA struct+lib for most of the selected species, except for *O. sativa* and *Z. mays*. TEtrimmer results based on RepeatModeler2 (RM2+TEtrimmer in Fig. 6) generated 1.2-to 2.2-times more Perfect families and consistently identified more DNA and LINE elements as Perfect families across all selected organisms and the three simulated genomes compared to RepeatModeler2 alone (RM2 in Fig. 6). Conversely, fewer LTR retrotransposons were identified by TEtrimmer for *D. rerio*, *O. sativa*, and *Z. mays* (Fig. 7 and Source Data file), which might be due to the relatively inefficient identification of low-copy TEs by TEtrimmer. Overall, TEtrimmer predominantly improved the curation of TEs in all selected organismal and simulated genomes, except for LTR retrotransposons in the two plant genomes.

## TEtrimmer exhibits improved genome-wide TE annotation ability in comparison to EDTA and RepeatModeler2

In addition to the direct comparison of TE consensus libraries, we benchmarked EDTA, RepeatModeler2, and TEtrimmer concerning whole-genome TE annotation by using a confusion matrix[34,45]. First, all genome regions annotated as any type of TE, termed all TEs, were analyzed to assess the accuracy of TE masking in the genome. TEtrimmer exhibited higher sensitivity and precision (F1 scores >0.941) than EDTA and RepeatModeler2 for *B. hordei*, *D. melanogaster*, *D. rerio*, *Z. mays*, and three simulated genomes, except for a slightly lower sensitivity score for the *D. rerio* EDTA+TEtrimmer library

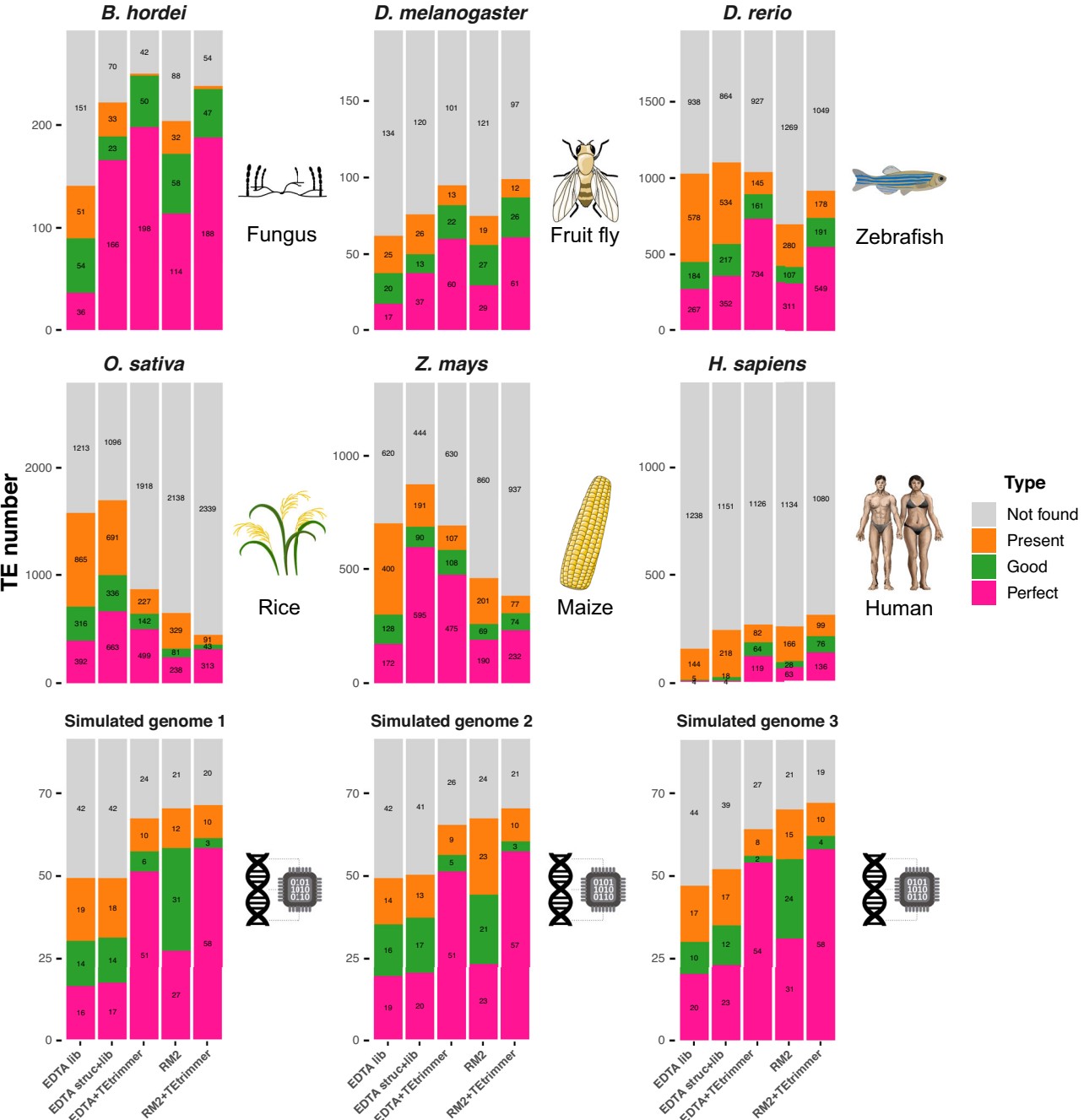

**Fig. 6 | TEtrimmer improves the discovery of intact TEs compared to EDTA and RepeatModeler2.** We benchmarked the performance of EDTA, RepeatModeler2 (RM2), and both these tools after additional TEtrimmer analysis (EDTA+TEtrimmer and RM2+TEtrimmer, respectively) by comparing the TE quality in the de novo-generated consensus libraries with the quality in the corresponding reference TE consensus libraries (Table 1) of the genomes of six organisms, i.e., *B. hordei*, *D. melanogaster*, *D. rerio*, *O. sativa*, *Z. mays*, and *H. sapiens*, and three simulated genomes containing 50 Mb of sequence and 50% TEs (bottom left), 100 Mb and 50% TEs (bottom middle), and 100 Mb with 75% TEs (bottom right). Two libraries were used for EDTA, termed EDTA lib and EDTA struct+lib. EDTA lib represents the EDTA consensus library, and EDTA struct+lib was generated by combining the EDTA consensus library with all TE sequences discovered by the EDTA structure-based module. The stacked bar graphs show the proportion of correctly discovered consensus TEs by the respective analysis tools/pipelines indicated on the *x*-axis according to the color code shown on the right. The *y*-axis displays the number of TE sequences in the respective reference TE library (sum of the values in the stacked bar graphs), i.e., 292 for *B. hordei*, 196 for *D. melanogaster*, 1967 for *D. rerio*,

2786 for *O. sativa*, 1320 for *Z. mays*, 1391 for *H. sapiens*, and 91 for the simulated genomes. Benchmarking categories included Perfect (pink), Good (green), Present (orange), and Not found (grey)[33]. The remaining TE sequences were assigned to the Not found category. Third-party icons used in this Figure: rice_cartoon icon by Daniel Carvalho (https://figshare.com/authors/Plant_Illustrations/3773596), human_female / Human_male icon (DBCLS https://togotv.dbcls.jp/en/pics.html), and zebrafish_simplified icon by DBCLS are licensed under CC-BY 4.0 Unported (https://creativecommons.org/licenses/by/4.0/); fruitfly_drosophila-yellow icon by Servier (https://smart.servier.com/) and corn icon by Servier (https://smart.servier.com/) are licensed under CC-BY 3.0 Unported (https://creativecommons.org/licenses/by/3.0/). The icon associated with the simulated genomes was generated by combining a DNA helix icon by DBCLS, licensed under CC-BY 4.0 Unported (https://creativecommons.org/licenses/by/4.0/), and a CPU icon downloaded from https://publicdomainvectors.org (https://publicdomainvectors.org/en/free-clipart/Digital-signal-processor-vector-illustration/13458.html), licensed under CC0 1.0 Universal (https://creativecommons.org/publicdomain/zero/1.0/). Raw data: Source Data file.

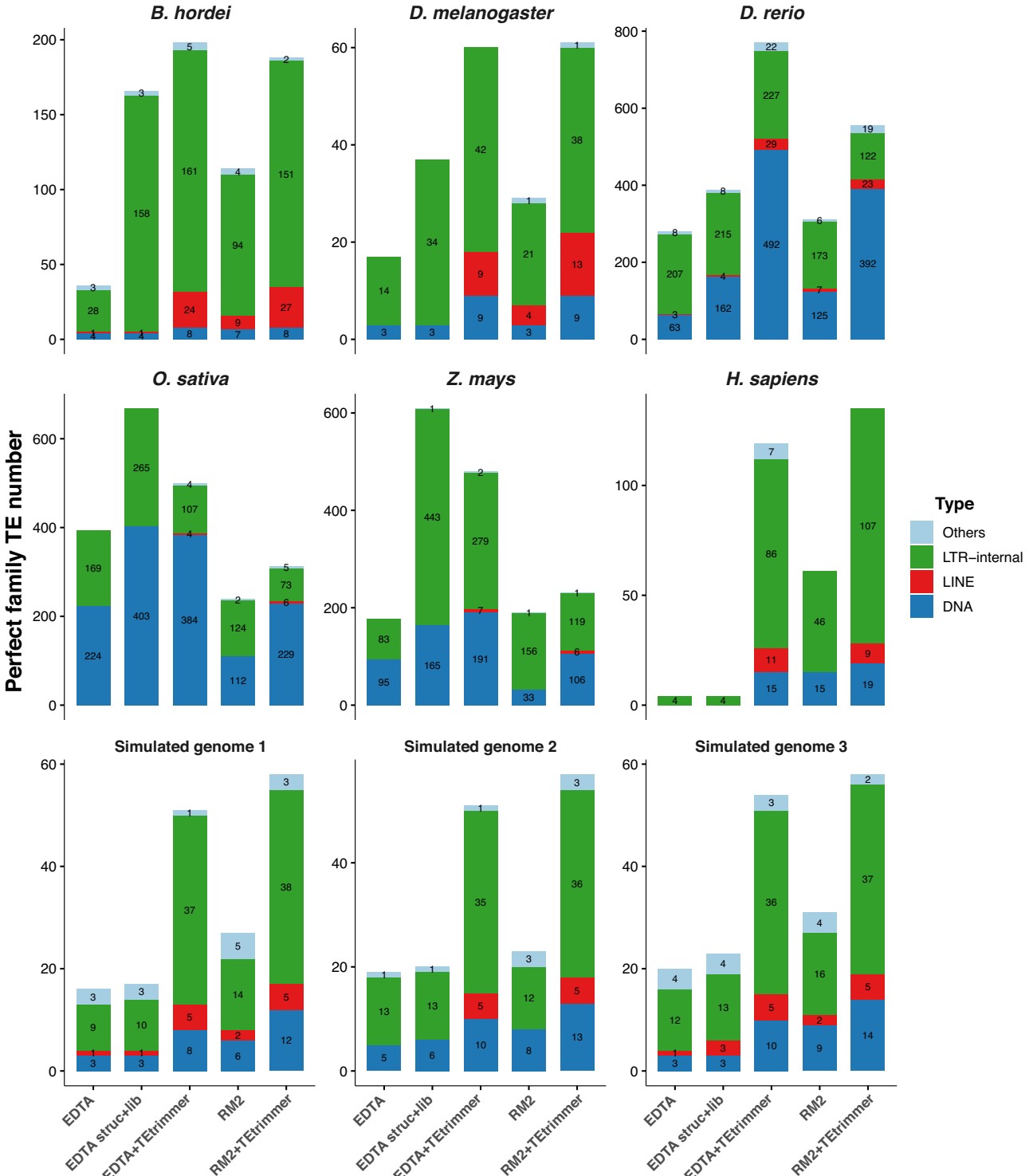

**Fig. 7 | LTR retrotransposon internal sequences and DNA transposons are predominant among the discovered intact TEs.** The stacked bar graphs show the number of Perfect matches from Fig. 6, resolved for the TE types of DNA transposons (blue), LINEs (red), LTR retrotransposon internal sequences (LTR-internal, green), and others (including e.g. Helitrons, SINEs, and simple repeat elements; light blue). The x-axis displays the tools/pipelines, and the y-axis shows the number of TE sequences. Raw data: Source Data file.

compared to EDTA alone (Fig. 8A and Supplementary Fig. 5). By contrast, for *O. sativa* (rice), TEtrimmer exhibited lower sensitivity but higher precision than EDTA and RepeatModeler2. TEtrimmer also showed lower sensitivity and precision for *H. sapiens* compared to the other tools (Fig. 8A and Source Data file).

Due to the overall poor performance on TE consensus library construction for *H. sapiens* by all tested tools (Fig. 6) and

TEtrimmer's relatively low TE annotation score for *H. sapiens* (Fig. 8A), we calculated a confusion matrix for different TE types (radar plots in Fig. 8B). In the case of LTR retrotransposons, LINEs, SINEs, and DNA transposons, both the sensitivity and precision were improved by additional TEtrimmer analysis. Moreover, genome-wide TE annotation based on the TEtrimmer consensus libraries exhibited better agreement with the reference annotation than those from

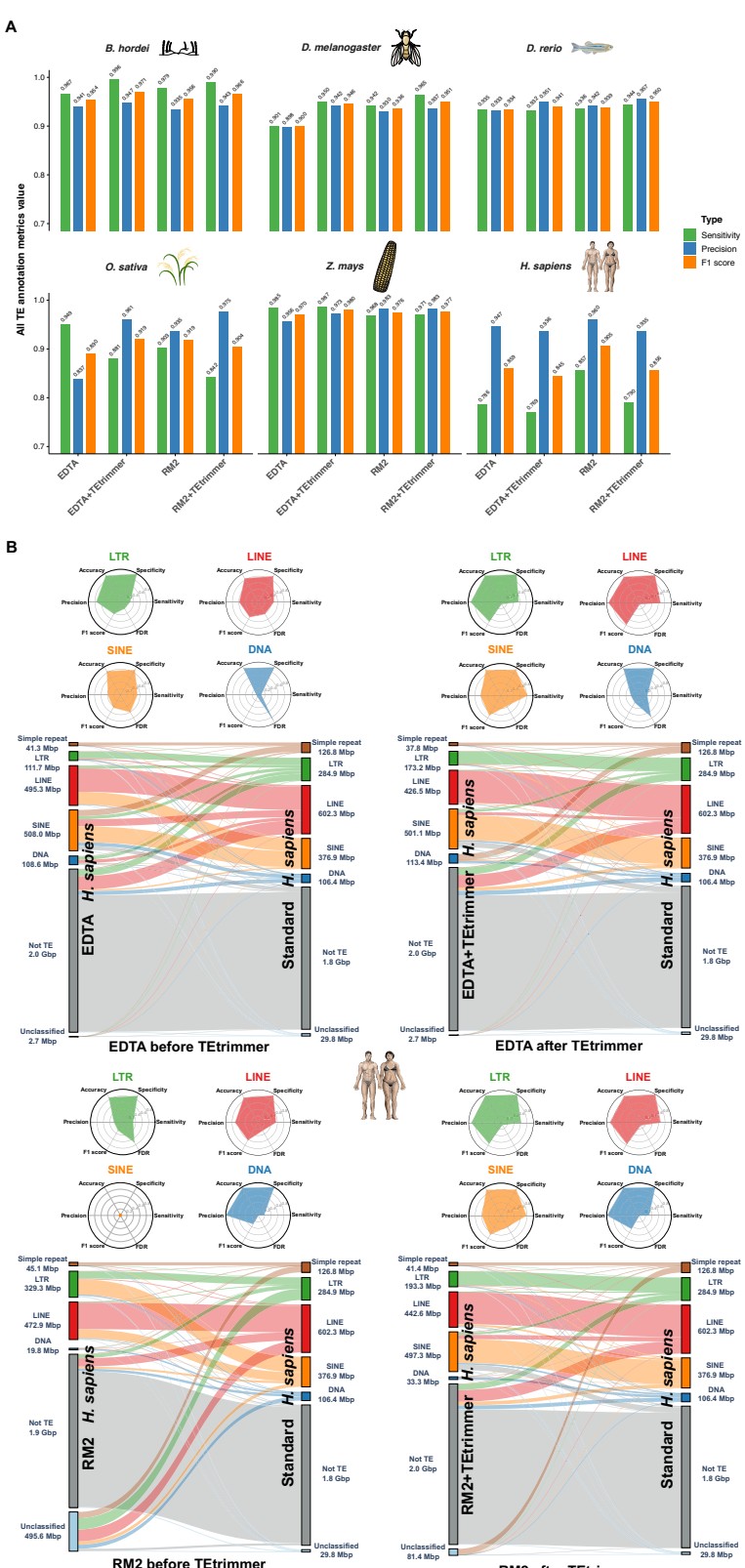

EDTA and RepeatModeler2 alone (Sankey plots in Fig. 8B). For instance, the TE annotation from RepeatModeler2 (before TEtrimmer) yielded 495.6 Mbp of unclassified TEs and no SINE elements for *H. sapiens*, while additional TEtrimmer analysis reduced the unclassified TE category to 81.4 Mbp and annotated 497.3 Mbp of SINEs with high accuracy (Fig. 8B). In conclusion, TEtrimmer libraries generally outperformed EDTA and RepeatModeler2-derived libraries

for whole-genome TE annotation across most of the tested organisms. Additionally, TEtrimmer provided more correct TE classifications.

**TEtrimmer is a user-friendly and computationally efficient tool**

TEtrimmer provides flexible installation options, including all dependencies, such as Conda, Docker, and Singularity containers, enabling

**Fig. 8 | TEtrimmer exhibits improved genome-wide TE annotation ability in comparison to EDTA and RepeatModeler2.** We assessed the genome TE annotation performance of EDTA, RepeatModeler2 (RM2), and both after additional TEtrimmer analysis (EDTA+TEtrimmer and RM2+TEtrimmer, respectively). All genome-wide TE annotation results based on de novo-generated libraries were compared with the results of the respective TE reference libraries (Table 1). We used a confusion matrix[34,45] to calculate the sensitivity, precision, accuracy, specificity, F1 score, and false discovery rate (FDR) of the tools. The genomes of six organisms were used for the analysis, i.e., *B. hordei*, *D. melanogaster*, *D. rerio*, *O. sativa*, *Z. mays*, and *H. sapiens*. **A** The bar plots show the sensitivity (green), precision (blue), and F1 score (orange) calculated with a confusion matrix for the six organisms indicated on top of each graph. The *x*-axis represents the respective analysis tools, and the *y*-axis displays the metrics score values based on overall genome TE annotation correctness (All TEs) at a range of 0.7 to 1.0. **B** Detailed genome-wide TE annotation benchmarking for *H. sapiens*. The radar plots in the upper panels show the benchmarking metrics for LTR retrotransposon (green),

LINEs (red), and SINEs (orange). The lower panels display the annotation overlap and differences between de novo-generated TE libraries from the indicated tool (left) and the reference consensus TE library for *H. sapiens* (right); each bar indicates the number of Mbp masked as the respective element in the genome. The links between the left and right bars indicate the connections between the respective libraries. TE types shown are LTR retrotransposon (green), LINE (red), SINE (orange), DNA transposon (dark blue), unclassified (light blue), and genomic sequence not identified as TE (Not TE; grey). Third-party icons used in this Figure: rice_cartoon icon by Daniel Carvalho (https://figshare.com/authors/Plant_Illustrations/3773596), human_female / Human_male icon (DBCLS https://togotv.dbcls.jp/en/pics.html), and zebrafish_simplified icon by DBCLS are licensed under CC-BY 4.0 Unported (https://creativecommons.org/licenses/by/4.0/); fruitfly_drosophila-yellow icon by Servier (https://smart.servier.com/) and corn icon by Servier (https://smart.servier.com/) are licensed under CC-BY 3.0 Unported (https://creativecommons.org/licenses/by/3.0/). Raw data: Source Data file.

## Table 2 | Runtime test of TEtrimmer for the genomes of four organisms

| Species | Genome size (Mbp) | EDTA as input for TEtrimmer | | | RepeatModeler2 as input for TEtrimmer | | |
|---|---|---|---|---|---|---|---|
| | | Input TE number | Runtime (h)¹ | Output folder size (GB) | Input TE number | Runtime (h) | Output folder size (GB) |
| *B. hordei* | 124 | 996 | 0.92 ± 0.049 | 2.30 ± 0 | 818 | 0.83 ± 0.040 | 2.10 ± 0 |
| *D. melanogaster* | 144 | 819 | 0.66 ± 0.067 | 0.92 ± 0 | 480 | 0.66 ± 0.046 | 0.96 ± 0 |
| *D. rerio* | 1679 | 8631 | 4.95 ± 0.225 | 15.10 ± 0 | 3504 | 2.32 ± 0.066 | 5.50 ± 0 |
| *O. sativa* | 373 | 10,404 | 3.30 ± 0.200 | 7.30 ± 0 | 2334 | 1.31 ± 0.090 | 2.50 ± 0 |

¹TEtrimmer was executed three times for each TE library with 48 CPUs and 140 GB RAM. The runtime averages and standard deviations are shown.

easy deployment across various computing environments, particularly high-performance computing (HPC) platforms. To help users quickly understand TEtrimmer's functionality, three comprehensive tutorial videos are available at (https://tehub.org/en/Tutorials), i.e., (1) TEtrimmer Background Introduction, (2) Step-by-Step Guide to the TEtrimmer GUI, and (3) Introduction to TEtrimmer Parameters. These resources provide clear explanations of the underlying principles, step-by-step guidance on the GUI, and detailed instructions for adjusting parameters to accommodate specific analytical requirements.

We evaluated the computational efficiency of TEtrimmer by testing the computation time of TEtrimmer with genomes from four organisms, *D. melanogaster*, *D. rerio*, *O. sativa*, and *B. hordei*, utilizing 48 CPU cores under default parameters (see "Methods" for details). TEtrimmer demonstrated rapid runtime performance by completing analysis within less than six hours in each of these cases (Table 2). Notably, TEtrimmer required only ~0.7 h to process the *D. melanogaster* genome when inputs derived from either EDTA or RepeatModeler2 were used. The longest processing time observed was about 5 h for the *D. rerio* genome using EDTA inputs, while utilizing RepeatModeler2-generated libraries as input reduced the runtime to around 2.3 h. In general, inputs from RepeatModeler2 resulted in shorter runtimes and smaller output folder sizes compared to EDTA, except for *D. melanogaster* where runtimes were comparable.

## Discussion

Due to decades of genome sequencing and the advanced long-read third-generation sequencing technology, the number of high-quality genomes is growing rapidly. However, precise and accurate TE annotation in genomes remains a major challenge. Several tools have been developed to automate TE discovery and annotation. Nonetheless, the manual curation of TEs is still mandatory to obtain comprehensive and detailed insights into the TE landscape of a given genome. Manual curation is a time-consuming and challenging process, and often beyond the scope and capacities of a research project. In this study, we introduce TEtrimmer, a software designed to automate and support

the manual curation of TEs. TEtrimmer efficiently clusters and cleans MSAs and precisely defines TE boundaries. Moreover, TEtrimmer provides comprehensive report plots and a user-friendly GUI application to facilitate the inspection and improvement of discovered TE consensus sequences. In particular, the sliding window-based cleaning strategy of the MSA, which results in high precision, and the interactive GUI application are distinguishing features of TEtrimmer as compared to other related software tools.

In manual curation procedures, the first labor-intensive step is separating the MSA into different clusters according to the alignment pattern and TE sequence similarity (Fig. 2). To accelerate this step, we combined the selection of distinct MSA columns, a phylogenetic tree approach, and a machine learning method (DBSCAN) to cluster MSAs automatically and accurately. Although other clustering tools exist, none of these are fully automated. For example, TreeCluster can group phylogenetic trees according to the relative branch distance[48], but the final cluster number must be defined manually. Furthermore, CD-HIT-EST can cluster TE sequences without generating an MSA[43], but the identity threshold needs to be manually adjusted for each MSA separately. MCHelper[39] also provides an MSA clustering function and, like TEtrimmer, relies on DBSCAN. However, MCHelper does not include the selection of divergent MSA columns and lacks the phylogenetic tree approach. It instead uses the K2P (Kimura 2-Parameter) sequence distance matrix for DBSCAN analysis. Selectively using divergent MSA columns for clustering is important to reduce the runtimes of phylogenetic tree construction. We found that phylogenetic trees are helpful to calculate sequence distances for processing by DBSCAN. In contrast to the tools mentioned above, the TEtrimmer clustering algorithm automatically defines the appropriate cluster number and thresholds. It combines the selection of divergent MSA columns and a phylogenetic tree strategy for a more advanced DBSCAN clustering analysis. This approach is not only useful for the clustering of TE sequences but also for any set of nucleotide or amino acid sequences. However, because the clustering module of TEtrimmer relies on MSAs, it cannot handle large-scale datasets containing thousands of DNA sequences.

Besides MSA clustering, we developed a series of MSA cleaning algorithms to remove gappy columns and highly divergent regions in MSAs (Fig. 3). Removing deviating sites from MSAs is important because they can hinder the generation of TE consensus sequences and the definition of TE boundaries. Fragmented TEs and excessive end extension during the curation process typically result in the addition of many TE-unrelated flanking sequences in the MSA. This issue is especially pronounced in the case of the 5' ends of LINE elements, which are frequently truncated[13] (Fig. 3). Other tools, such as trimAl[49], ClipKIT[50], and CIAlign[51], can perform MSA cleaning but are mainly designed to remove gappy columns and optimize MSAs for better phylogenetic tree construction. CIAlign provides basic functions to clean rows and columns of MSAs but mainly relies on alignment gap information, which is insufficient for TE-related MSA scenarios. TEtrimmer introduces a sliding window-based strategy to analyze sequentially the entire MSA nucleotide by nucleotide. Our benchmarking of the cleaning performance and accuracy (Fig. 3C) demonstrates that the TEtrimmer MSA cleaning module exhibits a performance close to that of manual MSA cleaning. In addition, we provide a GUI to enable users to revise sequences in the TE library, which we recommend in particular for TE types such as LINEs and SINEs due to their frequently truncated 5' ends. In summary, TEtrimmer is to the best of our knowledge the only tool that provides extensive automated MSA cleaning procedures for TE sequences.

TEtrimmer provides a powerful GUI application to facilitate the inspection and improvement of its outputs. This can close the gap between de novo-based TE libraries and the traditional, manual curation-based TE consensus libraries. TEtrimmer also introduces a TE discovery- and curation-related GUI application. Moreover, we offer more TE-related functions in this GUI application. For example, users can not only examine and improve TEtrimmer outputs but also any TE consensus libraries established by other tools such as EDTA and RepeatModeler2 (Supplementary Fig. 6). They can further easily perform BLASTN searches, MSA generation, MSA sequence extension, MSA cleaning, MSA plotting, TE-Aid style plotting, and consensus sequence generation. The integration of TE benchmarking functions into the GUI application is a future option.

To test the performance of TEtrimmer regarding the discovery of intact TEs, we executed TEtrimmer using the outputs from EDTA and RepeatModeler2, respectively, based on six organisms of different kingdoms of eukaryotic life and three simulated genomes (Table 2). The TE consensus libraries generated by TEtrimmer consistently contained a higher number of intact (Perfect) TEs than the consensus libraries generated by EDTA and RepeatModeler2 (Fig. 6). However, TEtrimmer discovered fewer intact LTR retrotransposons for *D. rerio*, *O. sativa*, and *Z. mays* in some cases. A possible explanation for this contrasting performance could be that TEtrimmer mainly relies on the repetitive nature of TEs to improve the consensus library and, in turn, may not be effective in accurately discovering low-copy TEs. In contrast to TEtrimmer, both EDTA and RepeatModeler2 rely on structure-based methods to identify low-copy LTR retrotransposons. Nevertheless, such low-copy TEs can be easily recovered by TEtrimmer with the provided GUI application through the optional review and curation process. While benchmarking based on organismal TE consensus libraries might be confounded by the absence of a ground truth, TE integration sites are fully known in the case of simulated genomes, rendering these ideal controls. They thus represent invaluable additions to the benchmarking process based on organismal genomes. Notably, TEtrimmer also improved the consensus libraries generated by EDTA and RepeatModeler2 for three simulated genomes of different sizes and TE content (Table 2), especially by increasing the number of intact (Perfect) TEs (Fig. 6), particularly DNA transposons and LTR retrotransposons (Fig. 7).

We noted, however, that none of the tested approaches (EDTA, RepeatModeler2, or TEtrimmer) consistently recovered more than 50% of the TE consensus libraries across benchmarked genomes, and performance varies widely (Fig. 6). Structure-based methods, including EDTA, are highly effective in detecting TEs with well-defined structural features, such as LTRs or TIRs, but struggle with elements lacking distinct structural motifs, like LINEs and SINEs[34,52,53]. On the other hand, repeat-based methods like RepeatModeler2, RepeatScout, and RECON rely on the presence of multiple similar copies to generate consensus sequences and tend to underperform in finding low-copy or highly diverged TEs. TEtrimmer also relies on the repetitive nature of TE sequences to construct high-quality consensus sequences, while the provided GUI enables users to recover low-copy or highly diverged TEs.

In addition to the comparative assessment of TE integrity, we introduced a confusion matrix[34] to evaluate the TE annotation performance at a genome-wide scale. EDTA and RepeatModeler2 exhibited great sensitivity and precision for the selected organisms at the level of all TEs, which indicates that these tools can accurately mask TE regions in the respective genomes. However, most of the benchmarking metrics were improved after TEtrimmer analysis, except for *O. sativa* and *H. sapiens* (Fig. 8A), where the metrics values were slightly decreased, most likely due to the relatively poor performance of TEtrimmer regarding the analysis of low-copy TEs. Nevertheless, we found an overall improved TE classification accuracy for *H. sapiens* by TEtrimmer (Fig. 8B). We also developed a script to allow users to benchmark TE annotations easily (https://github.com/qjiangzhao/TEtrimmer/blob/main/tetrimmer/TE_anno_benchmarker.py).

To date, EarlGrey and MCHelper are the two most prominent tools designed for automating the manual curation of TEs. TEtrimmer outperforms these tools in six main aspects. (1) EarlGrey does not provide a clustering function for MSAs, which can be a limitation for analyzing highly diverged TEs. Compared to MCHelper, TEtrimmer has a more sophisticated MSA clustering strategy that is based on the selection of divergent MSA columns in combination with the generation of a phylogenetic tree, which can dramatically improve DBSCAN performance. (2) Both EarlGrey and MCHelper are not able to clean MSAs efficiently, due to a lack of functions to clean MSA rows. (3) For each iteration of MSA extension, EarlGrey and MCHelper extend all TE sequences in the MSA. By contrast, we developed an MSA extension algorithm that can selectively extend TE sequences based on the divergence in the alignment. This strategy facilitates even the analysis of frequently truncated regions, such as the 5' end of LINE elements. (4) TEtrimmer assigns different evaluation levels for each output based on criteria such as the presence of terminal repeats, classification status, MSA sequence number, number of full-length BLASTN hits, and PFAM domain prediction (Supplementary Table 1). This helps in systematically categorizing and assessing the quality of the outputs. (5) TEtrimmer provides comprehensive report plots (Fig. 4) and a user-friendly GUI application (Fig. 5), which allows users to inspect and improve each TEtrimmer output sequence conveniently. We also integrated the MSA cleaning, MSA extension, and plotting functions into the GUI application to achieve manual curation-level TE libraries efficiently. (6) TEtrimmer is a computationally efficient tool. Compared to MCHelper, TEtrimmer required less than 3 h for analyzing TEs from the *D. rerio* genome when the RepeatModeler2-generated TE library was used as input (Table 2), while MCHelper needed ~30 h under a similar setup[39].

In conclusion, TEtrimmer is a powerful and user-friendly tool designed to automate the manual curation of TEs, addressing the major challenge of accurately annotating these repetitive DNA sequences in eukaryotic genomes. By integrating advanced strategies for MSA clustering, cleaning, and extension, as well as TE boundary definition, TEtrimmer enhances the quality of TE annotations markedly. The tool provides detailed report plots and a user-friendly GUI application for efficient inspection and refinement of results, making it accessible even to researchers who may not have extensive expertise in

TE genetics. Comprehensive benchmarking against the genomes of six diverse eukaryotic organisms demonstrated the advanced performance of TEtrimmer compared to existing tools like RepeatModeler2 and EDTA, particularly in identifying intact (full-length) TEs. Hence, TEtrimmer bridges the gap between automated TE annotation and the gold standard of manual curation, offering a robust solution for the accurate and efficient annotation of TEs in genomic studies.

## Methods

### BLASTN searches

TEtrimmer v1.5.1 (Qian et al., 2025; https://doi.org/10.5281/zenodo.16682752) uses the TE consensus library output from any de novo TE discovery tool, such as EDTA, RepeatModeler2, and REPET, or consensus TE libraries from closely related species, as input to improve the TE library quality. First, TEtrimmer removes duplicated input sequences via CD-HIT-EST[43] (v.4.8.1) when the `--dedup` option is enabled. Sequences with more than 95% sequence identity and coverage are merged into one element. Then, each input sequence is converted into a single file to enable multi-thread analysis. TEtrimmer uses these separated sequences to perform BLASTN searches against the respective genome to find query copies for each TE sequence with predefined parameters, `-evalue 1e-40 -max_target_seq 10000 -qcov_hsp_perc 15`. By default, only the 70 longest and 30 randomly selected BLASTN hits are used for further analysis when more than 100 target sequences are identified by the BLASTN query, as these are typically sufficient for meaningful downstream analysis. Finally, the BLASTN search result is converted into a BED file.

### MSA generation

TE sequences are extracted based on the BED file derived from the BLASTN search of each input TE consensus sequence against the corresponding genome (see above) by `bedtools getfasta`[54] (v2.31.1). Those sequences are aligned by MAFFT[55] (v7.520) with the fast alignment option `--retree 1` and any gappy alignment columns are removed before MSA clustering by the function `remove_gap_columns` (see section MSA cleaning below).

### MSA clustering

Because of the typically low sequence divergence among closely related TEs, several types of TE variants are usually included in the MSA based on the BLASTN search derived from each input consensus sequence against the respective genome (Fig. 2). To separate these variants, TEtrimmer calculates the percentage of each type of nucleotide in every column. Gaps in that column are not included in the calculation. Divergent columns are defined as those where all nucleotide proportions are lower than 0.8 in that column. In addition, TEtrimmer selects indel regions by identifying gap blocks via the function `select_gaps_block_with_similarity_check`. A gap block is defined as a contiguous stretch of MSA columns where (1) the difference in gap proportions between adjacent columns within the block does not exceed 10%, and (2) the proportion of the dominant nucleotide at each boundary of the gap block must be larger than 80%, ensuring the gap block is flanked by conserved nucleotides (Supplementary Fig. 7).

Next, all divergent columns and gap blocks are merged into a new FASTA file, yielding an MSA of pseudo-sequences representing the divergent regions in the original MSA, which is used by IQ-TREE[56] to generate a maximum likelihood phylogenetic tree. We use the fixed IQ-TREE model $K2P + I$ to reduce the execution time of this module. Then, a sequence distance matrix is calculated based on the relative tree branch distances. Finally, the DBSCAN[44] machine learning method is performed based on the sequence distance matrix to cluster the sequences of the MSA, with epsilon set to 0.1 and minimum samples set to 2. More than five clusters can be generated after MSA clustering. By default, TEtrimmer only proceeds with the top five clusters containing the highest number of sequences, which provides, in our

hands, typically a good balance between sensitivity and computational intensity. Users can increase this number by the option `--max_cluster_num`. For older TEs, most of the sequences in the MSA may sort into the Noise cluster (Supplementary Fig. 1D). In cases where the Noise cluster contains more than 15 sequences, represents over 60% of the total MSA, and the number of normal clusters is below the specified `--max_cluster_num` threshold, the Noise cluster is used for all subsequent analyses like MSA cleaning and TE boundary definition.

### MSA cleaning (trimming)

For cleaning columns within an MSA, TEtrimmer uses the function `remove_gap_columns`. By default, TEtrimmer removes columns when the gap comprises more than 80% of the column or the nucleotide number in this column is less than five. In a subsequent step, columns with a gap percentage between 40% and 80% and where the predominant nucleotide constitutes less than 70% of the total nucleotides are deleted.

In addition to gappy columns, highly divergent sequence regions might occur in the MSA. TEtrimmer includes two functions to remove divergent regions from the ends of the MSA row by row. The function `crop_end_by_divergence` first calculates the proportion of each nucleotide in every column that has at least five nucleotides. Gaps are not included in this calculation. All nucleotide proportions in columns with fewer than five nucleotides are considered as zero. Then the entire MSA is converted into a proportion matrix by replacing each nucleotide with the corresponding proportion value, and all gap positions are converted to zero. For each row of the proportion matrix, two sliding windows (default size 40 bp) are created at both ends that move stepwise towards the opposite end of the sequence, nucleotide position by nucleotide position. Each sliding window stops when the mean of the nucleotide proportions within the sliding window is greater than the set threshold (default 0.7), indicative of having reached a conserved region. For each sliding window starting from the left end of the sequence, all nucleotide positions between the left end of the sequence and the stopping point (left boundary) of this sliding window are deleted. Similarly, for each sliding window that starts from the right end of the sequence, all nucleotide positions between the right end of the sequence and the stopping point (right boundary) of the sliding window are deleted. After completion of all rows, the process is repeated with a smaller sliding window (default size 4 bp) and a higher threshold (default 1.0) for fine-polishing the MSA ends.

The second function used to clean the MSA is called `crop_end_by_gap`. It deploys a similar sliding window cleaning principle as `crop_end_by_divergence`, but the difference is that `crop_end_by_gap` will not convert the MSA into a proportion matrix. Instead, it takes advantage of the original MSA. For each sequence in the MSA, two sliding windows (default size 250 bp) are created at both ends that stepwise move towards the opposite end of the sequence, nucleotide position by nucleotide position. Each sliding window stops when the proportion of gaps included in the sliding window is below the set threshold (default 0.1), indicative of having reached a conserved region. For each sliding window that starts from the left end of the sequence, all nucleotide positions between the left end of the sequence and the stopping point (left boundary) of this sliding window are deleted. Similarly, for each sliding window that starts from the right end of the sequence, all nucleotide positions between the right end of the sequence and the stopping point (right boundary) of the sliding window are deleted. The `crop_end_by_gap` function is also used to facilitate MSA extensions (see below).

### MSA sequence extension

Annotated TEs derived from de novo TE discovery software can be fragmented and truncated. To complete such TE sequences, TEtrimmer iteratively extends both ends of the sequences from the clustered MSA separately via `bedtools slop`[54]. The default step size is 1000 bp

for each end per extension, with a maximum total extension size of 7000 bp at each end of the sequences.

First, TEtrimmer performs left-end sequence extension of the MSA. To save computing time, after each round of sequence extension, TEtrimmer only aligns the newly extended sequences from the current round, followed by cleaning of gappy columns via the function `remove_gap_columns` as described above (see section MSA cleaning above). Then, the function `crop_end_by_gap` (see section MSA cleaning above) is employed to clean the MSA additionally row by row, and the sum of the removed sites for each sequence (row) is recorded. The sequence is excluded from the next round of extension when the sum of its removed sites by `crop_end_by_gap` is more than 90% of the sequence length, indicating a sufficient extension for this sequence, as seemingly flanking genomic regions have been reached. Moreover, to assess if the overall left end extension is sufficient to include the left TE boundary, the newly extended and cleaned MSA (only containing the sequences extended in the current round) is used to generate a consensus sequence where the letter N represents any ambiguous position. An ambiguous position is an MSA column site where the proportion of the predominant nucleotide is below 0.7 or the total nucleotide count is below five. Then, a sliding window (default size 150 bp) is created at the left end of the TE consensus sequence that moves stepwise towards the opposite end of the sequence, nucleotide position by nucleotide position. The sliding window stops when the proportion of the ambiguous letter N (as defined above) within the sliding window is less than the set threshold (default 0.3). If the stopping point (left boundary) of the sliding windows is more than 300 bp away from the left end of the consensus sequence, TEtrimmer regards the left end sequence extension as complete. Otherwise, a new round of extension is initiated.

When the left end sequence extension is finished, TEtrimmer conducts right-end sequence extension of the MSA. Similar to the steps described above, right-end sequence extension also only aligns extended sequences from the current round and generates the respective consensus sequence. In contrast to the left end, to judge if the overall right end extension is sufficient, a right end sliding window is created only for this consensus sequence. The sliding window stops when the proportion of the ambiguous letter N (as defined above) within the sliding window is less than the set threshold (default 0.3). If the stopping point (right boundary) of the sliding window is more than 300 bp away from the right end of the consensus sequence, the right end extension is complete.

Note that this way, extension sizes for the two ends of each sequence in the MSA may differ. After sequence extension at both ends, TEtrimmer calculates the new (extended) genome coordinates (locations) for each sequence. Finally, extended sequences are extracted from the corresponding genome. All sequences are aligned, and gappy columns are removed from the MSA by the function `remove_gap_columns` as described above (see section MSA cleaning above).

## TE boundary definition
Sequence extension helps to complete TE consensus sequences, but excessive extension can also add other, non-TE sequences (false-positives) at both ends of the MSA. TEtrimmer uses two strategies to identify TE boundaries and to remove such inappropriate regions. Based on the final MSA after the extension step, first, a consensus sequence is generated for each MSA, and N is used to represent any ambiguous column where the proportion of the predominant nucleotide is below 0.8 or the total count of nucleotides is less than five. Next, a self-BLASTN search (with a relatively relaxed e-value of 0.05) of the consensus sequence is conducted to identify potential terminal repeats of TEs, including LTRs and TIRs. Once a terminal repeat is found, the TE boundary is set at the distal position of the terminal repeat, and the MSA is cropped accordingly. If no terminal

repeat is found, TEtrimmer checks whether the TE classification includes the terms LINE or SINE. In such cases, the user-defined poly-patterns, like poly(A), poly(T), or microsatellite tails, are examined to determine the 3' end boundary of these elements. If neither terminal repeats nor poly-patterns are detected, TEtrimmer selects sliding windows (default size 150 bp) at both ends of the consensus sequence that stepwise move towards the opposite end of the sequence nucleotide position by nucleotide position until the proportion of the ambiguous letter N is below 0.2 within the sliding windows. Then, only the MSA columns between the stopping points of the two sliding windows (including the sliding windows) are kept. This procedure removes the majority of false-positive flanking sequences. Furthermore, the functions `crop_end_by_divergence` and `crop_end_by_gap` (see section MSA cleaning above) remove highly divergent regions from the MSA. Based on the defined TE boundaries, TEtrimmer generates the corresponding consensus sequence for further analysis.

## ORF and PFAM domain prediction
For autonomous TEs, the presence of characteristic protein domains can be used to assist TE classification[13] and to determine TE orientation. TEtrimmer adopts the PFAM database[42] to annotate protein domains within TEs. It either uses the `--pfam_dir` option to query a local PFAM library, or can download the current PFAM database automatically in the absence of a local PFAM instance. Thereafter, TEtrimmer uses the EMBOSS function `getorf`[57] to predict ORFs and `pfam_scan.pl` (e-value 1e-2)[42] to search for TE protein domains against the PFAM database. Typically, all predicted PFAM domains share the same orientation. TEtrimmer generates reverse complements of the TE consensus sequence and the MSA if all PFAM domains are found on the antisense strand of the consensus sequence. If TE-related PFAM domains are found on both strands, TEtrimmer sums the length of PFAM domains found on both strands to decide on the orientation of the TE consensus sequence via a majority decision.

## TE classification
TEtrimmer includes two TE classification methods. First, it adopts the RepeatModeler2 module `RepeatClassifier`[33] to categorize TEs. By default, TEtrimmer re-assigns processed TEs if the sequence extension size is larger than 4000 bp, as these may represent mis-categorized and/or fragmented TEs. Further, two options are associated with the first classification method, named `--classify_all` and `--classify_unknown`. TEtrimmer runs RepeatClassifier for each output if the `--classify_all` option is used and only re-classifies Unknown TEs via the `--classify_unknown` option. If RepeatClassifier assigns a new category, TEtrimmer will replace the old assignment with it; alternatively, the original classification is kept. The second method is performed when the initial analysis of all the input TE sequences is completed. TEtrimmer uses the final, classified output sequences as a database to re-assign unknown sequences via RepeatMasker.

## Output evaluation
Five different evaluation levels, including Perfect, Good, Reco_check (i.e., recommended check), Need_check, and Low_copy, are assigned to each TEtrimmer output. TEtrimmer evaluates TE consensus sequences based on the presence of terminal repeats, classification status, the number of sequences in the MSA, the number of full-length BLASTN hits, and PFAM protein domain predictions (Supplementary Table 1). A full-length BLASTN hit is defined as a sequence with more than 90% coverage and more than 85% identity with the respective query sequence (i.e., the TEtrimmer-processed consensus sequence). When the number of sequences within the MSA is less than ten, TEtrimmer examines if the input sequence contains terminal repeats and if the number of full-length BLASTN hits is greater than or equal to

two. If so, this input sequence is regarded as a low-copy element; otherwise, it is excluded from further analysis.

### TE consensus library de-duplication

All consensus sequences processed by TEtrimmer are stored in the file TEtrimmer_consensus.fasta, which potentially contains duplicated sequences. To reduce any putative redundancy, TEtrimmer performs two rounds of de-duplication by CD-HIT-EST[43]. For the first round, consensus sequences are grouped into one cluster if they share more than 90% sequence identity and the alignment coverage of the shorter sequence is more than 90%. Afterward, TEtrimmer selects all clusters that contain at least one Perfect or Good (Supplementary Table 1) consensus sequence. For each selected cluster, if Perfect candidates exist, the longest Perfect consensus sequence is selected as the representative. If only Good candidates are present, the longest Good consensus sequence is selected. The second de-duplication round extracts all sequences from the remaining clusters and uses these sequences again as input for CD-HIT-EST. Sequences are grouped into one cluster if they have more than 85% identity, and the alignment coverage for all sequences is greater than 80%. Only the longest sequence in each new cluster is selected for the de-duplicated TE consensus library. The de-duplicated TE consensus sequences are saved into the text file TEtrimmer_consensus_merged.fasta.

### Generation of report plots

Seven plots are generated for each output to help evaluate the TEtrimmer output and to compare the differences between the TEtrimmer-processed and input consensus sequences. The report plots file includes (1) an MSA ends plot, (2) a whole MSA plot, (3 and 4) TE-Aid[13] plots for both input and output sequences, (5 and 6) a PFAM plot for both input and output sequences, and (7) a dot plot comparing input and output sequences (Fig. 4). For the MSA ends plot, TEtrimmer extracts 50 columns before and after the identified TE boundaries (see section TE boundary definition above) from the MSA and artificially joins them by ten hyphen symbols, signifying the TE (Fig. 4A). Nucleotides are represented by different colors, and the nucleotide background is colored if the nucleotide proportion in the column is less than 40%. The whole MSA plot module is derived from the software CIAlign[51], and the nucleotides are represented by different colored bars (Fig. 4B). In contrast to the default CIAlign plot, TEtrimmer adds two arrows to indicate the TE boundaries (TE start and TE end) in the whole MSA plot. TEtrimmer also includes the TE-Aid package to help with plotting. All TEtrimmer output and input TE consensus sequences are used for TE-Aid analysis (Fig. 4C and D). Furthermore, the PFAM plots are generated based on ORF and PFAM domain prediction results, which are denoted as blue and red arrows, respectively (Fig. 4E). The last dot plot is generated by the EMBOSS tool dotmatcher[57]. The position of the consensus output sequence after analysis by TEtrimmer is plotted on the x-axis, and the position of the corresponding input TE consensus sequence is plotted on the y-axis of the dot plot (Fig. 4F). All seven report plots are merged into one file to be used for convenient output evaluation.

### Genomes and reference TE consensus libraries used for genome-wide benchmarking

Six eukaryotic genomes and three artificially simulated genomes were used to test the TE annotation performance of TEtrimmer at the level of TE consensus libraries and on a genome-wide scale. These species comprised *B. hordei* (barley powdery mildew fungus), *D. melanogaster* (fruit fly), *D. rerio* (zebrafish), *O. sativa* (rice), *Z. mays* (maize), and *H. sapiens* (human) (Table 1). The reference TE consensus library of *B. hordei* is based on in-house, manually curated TEs[58]. In the case of *D. melanogaster*, *D. rerio*, *O. sativa*, and *Z. mays*, Repbase 28.10[15] was used for generating the respective reference TE

consensus libraries. *H. sapiens* reference TE consensus library was extracted from Dfam[16]. Simulated genomes were created using the Python scripts random_sequence_TEs.py and random_nest_TEs.py provided on GitHub (https://github.com/IOB-Muenster/denovoTE-eval)[46]. Simulated genome sizes were set to 50 Mb (seq_length: 25000000) with 50% repetitive content, 100 Mb (seq_length: 50000000) with 50% repetitive content, and 100 Mb (seq_length: 25000000) with 75% repetitive content, respectively. For all simulated genomes, the seed was set to 42 with a GC-content of 42%. A random sub-sample of the BeetRepeatDB[47] was used as reference repeats (up to 5 elements per TE lineage; provided as repeats.fa on GitHub: (https://github.com/qjiangzhao/TEtrimmerPaperFile). All simulated genomes, including the GFF annotation files, are available on the same GitHub repository.

### Benchmarking of MSA cleaning performance

Due to the presence of fragmented TEs and/or excessive end extension, the extended MSAs typically contain regions of low sequence conservation, which can be efficiently cleaned by the TEtrimmer cleaning module. The MSA cleaning capability of TEtrimmer was evaluated quantitatively by a customized confusion matrix[45]. Three MSAs were used for each evaluation, including the original MSA before cleaning, the manually cleaned MSA, and the MSA cleaned by TEtrimmer. The manually cleaned MSA served as the reference in the benchmarking process.

Both manual cleaning and TEtrimmer-based cleaning did not delete the nucleotides in lowly conserved (poorly aligned) regions but converted them to gaps (-). Consequently, all three MSAs had the same number of columns and sequences. The order of the sequences within the MSA was also identical. Thereafter, each position in the three MSAs was analyzed, and corresponding true positive (TP) sites, false negative (FN) sites, false positive (FP) sites, and true negative (TN) sites values were calculated and summed into TP, FN, FP, and TN values (Supplementary Table 2). For example, if a site in the original MSA was occupied by any type of nucleotide, this site was counted as true positive when both manual and TEtrimmer-based cleaning converted this site to a gap (-). At the end, the TEtrimmer cleaning sensitivity (TP / (TP + FN)), precision (TP / (TP + FP)), and F1 score (2 * (sensitivity * precision) / (sensitivity + precision)) were calculated and used to evaluate the TEtrimmer cleaning performance.

### Benchmarking of genome-wide TE discovery and annotation

We benchmarked TEtrimmer, RepeatModeler2, and EDTA on the basis of the corresponding reference TE libraries (Table 1) by two methods. The first method directly compared the consensus sequences to assess the integrity of TE sequences, which aimed to determine how many TE sequences from the reference library can be recovered by the respective de novo-generated library. The other method compared genome-wide TE annotation by the different tools to evaluate if TEs in the genome were correctly masked. Six organisms and three simulated genomes were used for the benchmarking test, and the reference TE consensus libraries and genomes were downloaded and prepared (Table 1). RepeatModeler2[33] with parameters `-threads 50 -LTRStruct` and EDTA with parameters `--threads 50 --sensitive 1` were run to generate de novo TE libraries for all selected organisms. Thereafter, de novo-generated TE consensus libraries were used as the input for TEtrimmer analysis with parameters `--num_threads 50 --classify_all`. Finally, all TE consensus libraries generated before and after TEtrimmer analysis were used individually to annotate the corresponding genome by RepeatMasker v4.1.1 with parameters `-pa 50 -s -a -inv -gff`.

First, we performed benchmarking by direct comparison of the TE consensus libraries. We used the benchmarking method developed by Flynn and co-workers[33] for this purpose. All TE consensus libraries generated by EDTA, RepeatModeler2, and TEtrimmer were compared

with the corresponding reference TE consensus library (Table 1). For EDTA, we created an additional library for the benchmarking by combining the EDTA consensus library with all TE sequences identified by the EDTA structure-based module. This new EDTA library was then de-duplicated using CD-HIT-EST with the parameters `-c 0.9 -aL 0.9 -aS 0.9`, resulting in the new library named EDTA struct+lib. Due to the default separation of flanking LTRs from internal sequences (LTR-internal) in the case of LTR retrotransposon by tools like EDTA and RepeatModeler2, as well as in reference TE consensus libraries, a custom Python script was developed to separate flanking LTRs from the internal sequences (LTR-internal) for all TEtrimmer consensus libraries. The respective script was named "SeparateLTR.py", which can be downloaded from the TEtrimmer GitHub repository. To avoid the benchmarking bias caused by flanking LTRs, we did not include the flanking LTRs for benchmarking. Four evaluation levels were assigned to each TE sequence in the reference library, including Perfect, Good, Present, and Not found. Perfect means that the reference TE consensus sequence has one match in the de novo-generated library with >95% similarity and coverage. Good indicates that the reference TE consensus sequence has multiple overlapping matches in the de novo-generated library, each with >95% similarity and total coverage. Present is similar to Good, but the required minimum similarity and total coverage are decreased to 80%.

Next, we performed the benchmarking at the level of genome-wide TE annotation. We developed a custom Python script named TE_sankey_plotter.py (available from TEtrimmer GitHub repository), which employs a confusion matrix to evaluate the genome-wide TE annotation and to illustrate the outcome comparatively via Sankey plots. The details about how to apply a confusion matrix on the genome-wide TE annotation were described before[34]. Briefly, the RepeatMasker.out annotation files derived from the TEtrimmer, RepeatModeler2, and EDTA consensus libraries, respectively, were compared with the RepeatMasker.out file based on the reference TE consensus library. First, all.out files were sequentially converted into BED files with four columns (chromosome name, start position, end position, and TE type). Then, the overlapping TE regions were resolved by BEDtools v2.31.1[54] with parameters `-d 0 -c 4,4,2,3 -o collapse, distinct, collapse, collapse`. Finally, six metrics scores were calculated, i.e., sensitivity, specificity, accuracy, precision, F1 score, and false discovery rate, to evaluate the genome-wide TE annotation.

### Installation and runtime test of TEtrimmer

TEtrimmer is written in the Python3 language and can be conveniently installed via a Conda package or a Docker image at (https://github.com/qjiangzhao/TEtrimmer). TEtrimmer can be run on the operating systems Linux, MacOS, and a Windows subsystem for Linux (WSL). There are no specific hardware requirements for running TEtrimmer. TEtrimmer is a multi-thread tool and supports high-performance computing (HPC). Detailed tutorial videos are available at (https://tehub.org/en/Tutorials).

We evaluated the runtime performance of TEtrimmer on genomes of four organisms, i.e., *D. melanogaster*, *D. rerio*, *O. sativa*, and *B. hordei*. For each genome, the analysis was executed three times using a compute node allocated via SLURM with 48 CPU cores (Intel Xeon 8468 Sapphire) and 140 GB of RAM. Runtime and output size were recorded for each repetition, and the mean and standard deviation were calculated across the three runs.

### Statistics and reproducibility

No statistical method was used to predetermine sample size.

### Reporting summary

Further information on research design is available in the Nature Portfolio Reporting Summary linked to this article.

### Data availability

TEtrimmer is an open-source software under the license GPLv3. We used large language models, ChatGPT and GPT4, to facilitate the development of TEtrimmer. A detailed TEtrimmer manual is supplied at (https://github.com/qjiangzhao/TEtrimmer) and a video tutorial at (https://tehub.org/en/Tutorials). All benchmarking results and scripts can be found on GitHub at (https://github.com/qjiangzhao/TEtrimmerPaperFile). All third-party software included in the TEtrimmer package are published under open-source licenses, with the exception of TE-Aid, for which we have approval by the author Clément Goubert (University of Arizona, USA). Source data are provided with this paper and can be found in the Source data file. Source data are provided with this paper.

### Code availability

The TEtrimmer code (v1.5.1) can be downloaded from GitHub at (https://github.com/qjiangzhao/TEtrimmer) and is also available via Zenodo (Qian et al., 2025; https://doi.org/10.5281/zenodo.16682752).

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

## Acknowledgements

We would like to thank to Anna V. Protasio and Lera Emmanuelle for their valuable feedback on the TEtrimmer software. We acknowledge Xinyi Liu (TUM, Munich, Germany) pre-testing TEtrimmer. We are grateful to Clément Goubert (University of Arizona, USA), who allowed the addition of TE-Aid into the TEtrimmer package. The TEtrimmer development and benchmarking were supported by high-performance computing at RWTH Aachen University under project IDs rwth0146 and rwth1554. This study was funded by the Deutsche Forschungsgemeinschaft (DFG, German Research Foundation) project number 274444799 [grant 861/14–2 awarded to RP] in the context of the DFG-funded priority program SPP1819 Rapid evolutionary adaptation – potential and constraints. Additional funding was provided by the United States National Science Foundation (NSF) grant 2122944 to MCW, the Ohio State University (OSU) Enterprise for Research, Innovation, and Knowledge (ERIK) STEM Education Faculty Startup Awards and JobsOhio, and the OSU start-up

fund to SO, as well as the German Federal Ministry of Education and Research (call Epigenetics: Opportunities for Plant Research, grant 031B1221) to TH).

## Author contributions

RP, SK, and JQ conceived this study.JQ and HX developed the TEtrimmer software.SK performed proofreading of the TEtrimmer code.HX and JQ drafted the TEtrimmer manual.LM generated the simulated genomes.SO provided EDTA results and performed intensive de-bugging of TEtrimmer.JS conducted RepeatMasker analysis.LF supported the development of the TEtrimmer MSA cleaning procedure and the MSA clustering algorithm.JQ and SK wrote the first draft of the manuscript.JQ drafted the Figures.TH supervised work performed by LM and provided funding.MW supervised work performed by HX and provided funding.RP and SK supervised the work performed by JQ.RP provided funding. All authors reviewed the manuscript and contributed to the editing.

## Funding

## Competing interests

The authors declare they have neither financial nor non-financial competing interests.
