## [Peer Review file · Nature Communications]

TEtrimmer: a tool to automate the manual curation of transposable elements

Corresponding Author: Professor Ralph Panstruga

Version 0:

Reviewer comments:

Reviewer #1

(Remarks to the Author)

This seems like a useful tool for characterizing transposable elements (TEs). This is an important topic, because so many genomes are being sequenced: they contain many types of TE that profoundly affected their evolution. But accurate TE characterization is an unsolved problem, without manual effort that is infeasible for many genomes.

It's a bit worrying that TEtrimmer has so many ad hoc parameters, window sizes, and thresholds.

I have little experience with most of the alternative tools, which somewhat limits the confidence of my review.

Minor comments

Abstract: "innovatively" sounds desperate (also in the Discussion).

Abstract: hard to understand: "phylogenetic analysis and DBSCAN to cluster TE alignment patterns"

Page 3: "TEs have been identified in all studied eukaryotic species": not quite true (<https://doi.org/10.3390/genes13050887>).

Page 3: "they occupy a large proportion of many genomes": please make clear if these are live TEs or mutated fossils.

Not quite right: "TEs can regulate promoter and enhancer sequences". Rather, TEs can be/contain promoter and enhancer sequences.

Page 4: "five major steps: BLAST...": better not to say "BLAST" here, because that's a specific software tool, whereas any similar aligner could be used instead.

Page 6: "separates each set of input sequences into a single file": should be: puts each input sequence in a separate file?

Fig 1: "generate consensus sequence" is missing.

Not exactly right: LINE elements having "highly divergent 5' regions" (page 10) or "highly degraded" (page 22). Rather, they often have 5' truncations.

Fig 4 legend "y-axis gives the nucleotide position": should be x-axis.

Fig 4C: explain that it shows solo LTRs. Near the top-right of the right panel, some alignments systematically end at the start of the right-hand LTR: what is going on there? Also, why is there such variation in "Divergence to consensus"? I would expect the divergence to fluctuate slightly around a value (e.g. 5%) that depends on the TE's age.

Page 15: "highly identical" is bad English.

Page 22: cluster MSAs "precisely" is over-claiming (sounds like "perfectly").

Page 22: "consistent columns can potentially mask or dilute the differences among sequences": I think that's incorrect, it doesn't harm sequence-distance based analysis.

Page 22: "in our experience, an un-rooted maximum likelihood phylogenetic tree is superior": sounds plausible, but I'm suspicious of "in our experience": did the authors do a controlled test?

"we introduced for the first time Sankey plots to the TE field": that sounds really awfully desparate. Sankey plots are fashionable and widely (over-)used.

Page 25: "By default, TETrimmer only proceeds with the top two clusters": that worries me, because it contradicts the Fig 2 example with 13 clusters. Please clarify.

Fig 6 shows that all methods still do quite badly: it would be useful to analyze what the main reasons are.

(Remarks on code availability)

I have briefly looked at the code website, and it does seem to be available.

Reviewer #2

(Remarks to the Author)

The manuscript titled "TETrimmer: a novel tool to automate the manual curation of transposable elements" by Qian et al. describes a novel tool to automate some steps of the laborious manual curation process to lower the barrier of entry to high-quality TE curation and aid in improving the generation of gold-standard TE resources. The manuscript is well-written and clearly outlines the rationale, approach, and findings. This is a valuable contribution to the TE community, and I commend the authors for this effort.

I have a few comments and questions that I would like the authors to clarify before acceptance. Importantly, several claims throughout the manuscript require statistical comparisons for support, as they are currently lacking in some sections. Additionally, increased transparency in specific areas would help researchers assess the tool's usefulness for their needs.

Main Comments

- On line 185, it states that "These criteria might not efficiently evaluate TEs lacking terminal repeats such as LINEs and SINEs". Does this mean that a LINE or SINE will never be evaluated by TETrimmer as "perfect" or "good"? Why was the decision taken to evaluate outputs in this manner? Would it be possible to implement a switch so that evaluation accounts for TE classification? For example, if a consensus is classified as a LINE/SINE, could the tool prioritize checking for 3' tail characteristics instead of terminal repeats? Given the quality of the tool and the developers involved, I imagine this would be feasible to implement—though it could also be a future improvement rather than a requirement for initial publication. This addition would be particularly beneficial for non-specialists.

- Table 1: Only relatively high-copy-number TEs (≥ 30 sequences) are evaluated as "perfect.". However, copy number alone does not necessarily indicate a better consensus sequence. Was this threshold based on observations during testing? I can imagine that recently active TEs might lead to "perfect" evaluations due to the presence of many intact copies. To what extent do the authors observe "perfect" TE consensus sequences with MSAs of < 30 sequences? It would be valuable to discuss how TETrimmer performs in such cases. A brief justification in the Methods section or a short 1–2 sentence explanation of why these cutoffs were chosen would improve transparency.

- Figure 2 (subfamily resolution approach).

o (i) How effective is this approach for older, fragmented, or highly diverged TE families? Would clustering produce multiple

noise clusters or fail to form clusters altogether? In scenarios where BLAST-extracted copies are highly diverged but still generate BLAST hits, do the authors have recommendations?

o (ii) [Fig. 2 and lines 239-240] Does the clustering process incorporate the original consensus sequence during the initial alignment, or does it rely solely on genomic copies? I have encountered cases where the final consensus shifts toward an abundant repetitive sequence adjacent to the original consensus in repeat-rich genomes. Does TEtrimmer account for this possibility, particularly in the cleaning process mentioned in Line 242? If not, could such biases affect the final output?

o (iii) The default setting resolves two clusters (Lines 666-667), and the authors state that increasing this number can lead to false positives. However, Figure 2 depicts 13 clusters plus a noise cluster, which may mislead readers into assuming this is typical behavior. While I understand the example was chosen to showcase the tool's potential, it would be beneficial to clarify in the figure legend or text that the default behavior is not to resolve all possible clusters (and the reasons why). Additionally, are all 13 clusters in Figure 2 true TE subfamilies? How does TEtrimmer define these subfamilies?

- Lines 250-251 and 703-704. We often find LINEs where 1 or 2 copies are intact, with the majority being truncated. How does TEtrimmer handle these cases? Here, the gap percentage could be >90%. Does this lead to truncation of full-length LINEs? If not, what safeguards are in place to prevent this? These scenarios are important to highlight, and the authors could take the opportunity to provide their recommendations (in the methods, or perhaps in the GitHub repository).

- Lines 257-259: The manuscript presents a single example of a high-copy-number LINE, but this does not represent the typical TE curation challenges we often face. The authors mention that `crop_end_by_gap` is better suited for LINEs, but how does its performance compare to `crop_end_by_divergence`? Are both equally effective for LINEs? Could TEtrimmer use them dynamically without requiring the user to choose? Additionally, it would be helpful to include an example of a less ideal case (e.g., a LINE with few intact copies and many fragments) to illustrate tool performance in more realistic scenarios.

- Lines 330-331: The manuscript states that TEtrimmer removed 4,835 bp from the consensus, eliminating one ORF and fusing two others. Maintaining intact ORFs is crucial for TE classification, so this raises concerns. In Figure 4E, the gag ORF appears to be missing, yet intact LTR retrotransposons should have both gag and pol. Is this fusion justified? Does the final consensus match expected biological structures? This issue requires further validation and justification what the TEtrimmer output is expected.

- Lines 434, 443, 490: The manuscript makes qualitative claims (e.g., "considerably") without statistical support. These should be backed by appropriate statistical tests, which could be reported in the text and/or visualized in Figure 6. For example, are increases in perfect/good/present/not found TE families statistically significant? Simple statistical tests would improve transparency and add support to the findings.

- Figure 7: The claim that TEtrimmer provides "superior genome-wide TE annotation ability" requires statistical backing. Figure 7B indicates classification conflicts between tools and reference annotations. Has misclassification been significantly reduced? Statistical comparisons of annotation improvements (e.g., total annotated base pairs, classification accuracy) should be provided.

Minor Comments

- Line 67: The manuscript states, "LINEs and SINEs are defined by their genomic poly-A repeats." Is this always the case? The Wicker et al. (2007) classification mentions additional characteristics (e.g., AT-rich tails, tandem repeats, poly(T) tails). In my experience, tandem repeats are common in some taxa. Clarifying this would strengthen the manuscript. As a side note, does this have any impact on the structure-based modules of TEtrimmer? Does it only look for poly-A signals?

- Line 287: What are the cutoffs that define lowly conserved regions? These could be defined in the text.

- Lines 661-662: How are indels represented in the pseudo-sequences representing divergent regions? If my understanding is correct, gaps are not counted towards the divergence calculation. In this case, is it possible that an indel column could have a nucleotide proportion >0.8 but represent a true subfamily (as the indel is present in a subset of sequences and might have high identity among those sequences containing the indel). Does this mean this column containing an indel position will not be counted as a variable region? How would this impact the inference of subfamilies? Often in manual curation, these are the easier subfamilies to define, so it is important that indels are features that can help to define subfamilies.

(Remarks on code availability)

The code is well-commented and easy to follow. Descriptions are provided where appropriate, and data is reproducible. The README is well-structured and provides enough guidance for usage of the tool.

Reviewer #3

(Remarks to the Author)

Recommendation: Revision

Summary:

This manuscript introduces TETrimmer, a novel tool designed to automate manual transposable element (TE) curation. TETrimmer addresses key limitations in existing tools by enhancing multiple sequence alignment (MSA) clustering, sequence cleaning, and TE boundary definition. The tool features a graphical user interface (GUI) and report generation, improving both usability and annotation accuracy. The authors provide a quantitative comparison of TETrimmer's performance against existing methods, demonstrating that TETrimmer enhances TE annotation precision while reducing manual workload.

Overall, this manuscript presents a comprehensive and well-structured study on TE annotation and curation, effectively justifying the need for TETrimmer. I also explored the GitHub repository and the YouTube instruction video, and I appreciate the detailed documentation, which will be highly valuable to future users. Below, I outline several questions and comments that could help further improve the manuscript:

1. Justification for MSA-Based TE Classification. MSA sequence-to-sequence alignment based classification can be computationally expensive. Why is MSA preferred over other sequence clustering methods, such as Hidden Markov Models (HMMs) or machine learning-based approaches? What are the benefits to use MSA as the major algorithm in TETrimmer?

2. Could CD-HIT-EST miss TE subfamilies during de-duplication, particularly if the clustering threshold is too high or if subfamilies are highly similar? Some TE subfamilies diverge due to mutations and indels, and CD-HIT-EST may overlook rare or ancestral subfamilies. Additionally, fragmented TEs might be erroneously clustered into separate groups instead of being assigned to their correct subfamily. Can the authors comment on this issue?

3. LINE 157-160: Is it necessary to separate each sequence into an individual file for multi-threading? Creating a large number of small files may introduce I/O bottlenecks. The authors can consider using batch processing instead of file-based separation. It might be a better idea to use multi-threading in a Single BLASTN Job with the built-in multi-thread parameters; This allows BLASTN to internally distribute sequence queries across multiple threads without excessive file I/O overhead.

4. Did the authors benchmark computational resources (runtime, memory usage) for TETrimmer against other tools? Given that the TE annotation can be a computationally intensive task, users would benefit from knowing the expected runtime and memory requirements for different genome sizes. At a minimum, the authors could provide this information for model genomes tested in the study.

5. LINE 416-429 and table 2. The authors used different TE reference libraries for each genome. While I understand that a custom TE library may be necessary for *Blumeria hordei*, why was RepBase not used for the human genome? This could make the comparison more reliable. It is also mentioned in line 486-487 "Due to the overall poor performance on TE consensus library construction for *H. sapiens* by all tested tools (Fig. 6A) and TETrimmer's relatively lower TE annotation score for *H. sapiens*" Would it be possible to improve the performance for human genome by using the RepBase human TE library?

6. Did the authors evaluate TETrimmer's performance on solo LTRs and nested TEs? These elements are common in large genomes and they are challenging for annotation. If tested, could the authors provide a discussion on its effectiveness in these more complicated scenarios?

7. After generating TE consensus libraries, how was whole-genome TE annotation performed? Did the authors apply RepeatMasker for each TE library to conduct genome-wide TE annotation, or was an alternative approach used?

8. It is not clear to me what was the ground truth for benchmarking whole-genome TE annotation? As described in lines 474-485. In the method section, the authors mentioned that all '.out' files were compared with Repeatmasker file. How do the authors justify RepeatMasker annotations as "perfect" ground truth? Would it be feasible to use simulated data for validation? For example, inserting synthetic TE sequences into a TE-free artificial genome and comparing annotation results across tools would provide a controlled accuracy assessment. The simulated TE insertions are the real ground truth can be used for calculating the matrix.

9. The methods section should include detailed BLASTN parameters to help readers understand the query criteria applied in the first step of TE identification.

10. Does TETrimmer perform differently on autonomous vs. non-autonomous TEs? Some steps in TETrimmer seem tailored for autonomous TEs, which may make them easier to detect. However, non-autonomous TEs are also important to genome evolution. Could the authors clarify whether TETrimmer is equally effective for both categories?

In conclusion, the manuscript provides a valuable contribution to the field of TE annotation and curation. The authors effectively demonstrate the need for TETrimmer, and the provided documentation enhances its accessibility. However, addressing the above technical questions, benchmarking details, and methodological justifications would improve the manuscript's clarity.

(Remarks on code availability)

I have reviewed the code, explored the GitHub repository, and gone through the documentation and instructional video. The

authors have done a great job organizing the code. They provide a detailed README outlining the download and installation steps.

Version 1:

Reviewer comments:

Reviewer #1

(Remarks to the Author)

The authors have addressed my concerns well, and I do not need to re-review this again. I have just one more comment.

I find this part of the Background gratuitously confusing: "Transposable elements (TEs) are selfish repetitive DNA elements that can move... they occupy a large proportion of many genomes, such as around 45% of the human genome".

Non-expert readers (such as my younger self) are misled by such statements into thinking that 45% of the genome consists of intact, active TEs. The plain meaning of "transposable element" is an intact TE.

(Remarks on code availability)

Reviewer #2

(Remarks to the Author)

The authors have addressed my comments with thoughtful and well-formulated responses.

I am unable to assess the responses that in particular refer back to new supplementary figures/tables, as the updated ones seem to not have been provided. (e.g. the responses refer to supplementary figure 7 with multiple panels, but supplementary figure 7 in the supplementary materials provided is a single figure with the title "TEtrimmer utilizes gap information from MSAs to assist in clustering TE sequences.")

I am supportive of publication of this work, which shows promise in lowering the barrier for manual TE curation, which remains a difficult endeavour.

(Remarks on code availability)

Reviewer #3

(Remarks to the Author)

The authors have addressed all my comments and questions. The revisions have improved the clarity and quality of the manuscript. I believe the manuscript can now be accepted.

(Remarks on code availability)

I have reviewed the code and tutorial; the authors provided enough instructions on GitHub.

Version 2:

Reviewer comments:

Reviewer #2

(Remarks to the Author)

The authors have addressed all my remaining queries and I am supportive of publication. Congratulations to the authors for creating a nice tool to aid in improving TE curation!

(Remarks on code availability)

Code is well-structured and instructions on GitHub are clear.

REVIEWER COMMENTS

Reviewer #1 (Remarks to the Author):

This seems like a useful tool for characterizing transposable elements (TEs). This is an important topic, because so many genomes are being sequenced: they contain many types of TE that profoundly affected their evolution. But accurate TE characterization is an unsolved problem, without manual effort that is infeasible for many genomes.

We welcome the recognition of the importance of the topic by this reviewer.

Comment 1. It's a bit worrying that TETrimmer has so many ad hoc parameters, window sizes, and thresholds.

Many thanks for the feedback regarding the number of *ad hoc* parameters in TETrimmer. We believe that providing as many adjustable parameters as possible provides benefits concerning flexibility and individual requirements for diverse applications; however, we also appreciate that a large number of parameters may overwhelm most users. To balance flexibility and usability, we have established thoroughly tested default settings for all parameters, which have been optimized across a wide range of eukaryotic genomes to ensure robust performance without the need for modification in many cases. Moreover, to assist in understanding each parameter, we have created a new tutorial video named "Introduction to TETrimmer Parameters", which provides clear explanations of the functionality of each parameter. We are confident that this lowers the barrier for new users while offering advanced customization for experienced users. This tutorial video can be found here:

<https://youtu.be/8jp3j5FFf1w>

or here:

https://www.bilibili.com/video/BV18c59zpEZZ/?share_source=copy_web&vd_source=e97586c562998df25f9322fd7a2705e6

We also integrated this tutorial video to the TETrimmer GitHub page and on TE Hub: <https://tehub.org/en/Tutorials>

We refer to this tutorial in the revised manuscript (lines 590 and 1062).

Minor comments

Comment 2. Abstract: "innovatively" sounds desperate (also in the Discussion).

We deleted "innovatively" here (line 46) and in the Discussion (lines 622 and 626).

Comment 3. Abstract: hard to understand: "phylogenetic analysis and DBSCAN to cluster TE alignment patterns"

We modified this sentence ("... it first combines phylogenetic analysis and DBSCAN to cluster TE alignment patterns precisely") in the Abstract as follows: "... it first applies a phylogenetic tree analysis together with the machine learning method DBSCAN to cluster TE alignment patterns accurately" (lines 44-45).

Comment 4. Page 3: "TEs have been identified in all studied eukaryotic species": not quite true (<https://doi.org/10.3390/genes13050887>).

Many thanks for pointing this out. We changed the phrasing to “almost all” (line 59).

Comment 5. Page 3: "they occupy a large proportion of many genomes": please make clear if these are live TEs or mutated fossils.

We thank the reviewer for the comment and agree: “Live TEs” and “mutated TE fossils” are the two extremes of an age spectrum for TEs, rather than two absolute categories. Most identifiable TEs usually are somewhere within this spectrum, and no matter which end they stand on, if they are identifiable as such, we refer to them as TEs. In any case, they occupy proportions of the genome (see, for example, the following reference: Wells & Feschotte, 2020 DOI: 10.1146/annurev-genet-040620-022145).

Comment 6. Not quite right: "TEs can regulate promoter and enhancer sequences". Rather, TEs can be/contain promoter and enhancer sequences.

We changed the sentence to “... TEs can contain promoter and enhancer sequences to alter host gene expression” (line 64).

Comment 7. Page 4: "five major steps: BLAST...": better not to say "BLAST" here, because that's a specific software tool, whereas any similar aligner could be used instead.

We changed the sentence to “Searching similar sequences in the genome, e.g., via BLAST ...” (lines 133-134).

Comment 8. Page 6: "separates each set of input sequences into a single file": should be: puts each input sequence in a separate file?

We changed the sentence to “TEtrimmer saves each set of input sequences into a separate file” (line 173-174).

Comment 9. Fig 1: "generate consensus sequence" is missing.

We added “Consensus sequence generation” to the flowchart in **Figure 1**.

Comment 10. Not exactly right: LINE elements having "highly divergent 5' regions" (page 10) or "highly degraded" (page 22). Rather, they often have 5' truncations.

We converted “highly divergent” to “frequently truncated” in line 295. Besides, we changed “highly degraded” to “frequently truncated” in lines 652 and 726.

Comment 11. Fig 4 legend "y-axis gives the nucleotide position": should be x-axis.

We changed “y-axis” to “x-axis” in this figure legend (line 393).

Comment 12. Fig 4C: explain that it shows solo LTRs. Near the top-right of the right panel, some alignments systematically end at the start of the right-hand LTR: what is going on there? Also, why is there such variation in "Divergence to consensus"? I would expect the divergence to fluctuate slightly around a value (e.g. 5%) that depends on the TE's age.

We appreciate the reviewer's feedback and acknowledge their meticulous attention to detail. The plots in **Figure 4C** show all BLASTN hits of the consensus sequence and their divergence from it

before (left) and after (right) the analysis with TETrimmer. It is noticeable that many hits occurred in the regions of the LTRs (shorter hits at both ends of the retrotransposon). This indeed might be due to many solo-LTRs present in the genome. Solo-LTRs are remnants of complete LTR retrotransposons, which are commonly generated by homologous recombination between two long terminal repeats (LTRs) within the same TE or between very closely related TE copies. Another possible explanation for these BLASTN hits is the presence of a non-autonomous partner element (TRIM/LARD). Near the top right of the right panel, there is an accumulation of hits that show a divergence of over 20% and that do not cover the LTR region. These hits very likely belong to a closely related LTR retrotransposon family with an especially high similarity in the region of the reverse transcriptase and RNase H domains. It is important to note that these hits are not considered in the consensus calculation but only show up in the BLASTN results. The large variation in the 'Divergence to consensus' (displayed on the y-axis) results from the plot type, which is a visualization of a genome-wide BLASTN search. Complete elements with a low divergence of <1.5% over 90% of the sequence length are displayed in red and represent full-length genomic copies (before TETrimmer analysis: 1; after TETrimmer analysis: 77). All other hits represent either older, diverged, or fragmented copies or copies from closely related families (before TETrimmer analysis: 450; after TETrimmer analysis: 1356). The classic "80-80-80" definition (Wicker et al., 2007) would even allow for a divergence of 20% between two TEs to still be considered from the same family.

Comment 13. Page 15: "highly identical" is bad English.

Agreed! We changed the wording from "To group highly identical sequences" to "To group sequences with high similarity" (line 420).

Comment 14. Page 22: cluster MSAs "precisely" is over-claiming (sounds like "perfectly").

Agreed! We changed "precisely" to "accurately" in line 45 and line 627.

Comment 15. Page 22: "consistent columns can potentially mask or dilute the differences among sequences": I think that's incorrect, it doesn't harm sequence-distance based analysis.

We thank the reviewer for the clarification. We agree that consistent columns do not mask the differences among sequences. The original statement has been removed to avoid any misleading implication (lines 636-639). We now clarify that the selection of only divergent columns is intended solely to reduce computational runtime for the phylogenetic tree construction (line 636).

Comment 16. Page 22: "in our experience, an un-rooted maximum likelihood phylogenetic tree is superior": sounds plausible, but I'm suspicious of "in our experience": did the authors do a controlled test?

In the MSA clustering process, we only need the relative distance relationships between TE sequences from the multiple sequence alignment (MSA), and it is difficult to select an appropriate outgroup or ancestral reference TE sequence. For this reason, we used unrooted phylogenetic trees. We did not perform a benchmarking comparison for the different phylogenetic construction methods. Accordingly, we changed the corresponding text to "We found that phylogenetic tree construction is helpful to calculate sequence distances for processing by DBSCAN" (lines 638-640).

Comment 17. "we introduced for the first time Sankey plots to the TE field": that sounds really awfully desparate. Sankey plots are fashionable and widely (over-)used.

We changed the sentence to “Here, we also developed a script to allow users to benchmark TE annotation easily (https://github.com/qjiangzhao/TEtrimmer/blob/main/tetrimmer/TE_anno_benchmark.py)” (lines 711-713).

Comment 18. Page 25: "By default, TEtrimmer only proceeds with the top two clusters": that worries me, because it contradicts the Fig 2 example with 13 clusters. Please clarify.

Thank you for pointing out this important observation: Our initial concern was that increasing the `--max_cluster_num` parameter would significantly expand runtime and computational resource demands. However, based on our latest runtime tests (see **Reviewer-only Table 1**), we observed no substantial difference in runtime between `--max_cluster_num = 2` and `--max_cluster_num = 5`. Instead, we observed that setting `--max_cluster_num = 5` yielded higher-quality outputs. Therefore, we adjusted the default value of `--max_cluster_num` to 5. For the MSA clustering example (**Figure 2** in the manuscript), although 13 clusters were generated, only clusters 1 and 2 contained more than 10 sequences—the default value of `--min_seq_num`. As a result, only the top two clusters would be used for further analysis, even with `--max_cluster_num = 5`. In our analyses across genomes of various sizes and complexities, we rarely observed more than five clusters containing over 10 sequences each in an MSA. Nevertheless, users are free to modify the `--max_cluster_num` parameter. With the help of the parameter tutorial video, users can choose a suitable value according to their specific needs.

Reviewer-only Table 1: Runtime comparison of TEtrimmer with different `--max_cluster_num` values.¹

Species	Genome size	Input TE number	Runtime (h) when <code>--max_cluster_num = 2</code>	Runtime (h) when <code>--max_cluster_num = 5</code>
B. hordei	124 Mbp	818	0.83	0.91
D. melanogaster	144 Mbp	480	0.66	0.71
D. rerio	1,679 Mbp	3,504	2.32	2.64
O. sativa	373 Mbp	2,334	1.31	1.74

¹All runtime tests used RepeatModeler2 as input for TEtrimmer and are based on 48 CPUs and 140 GB RAM. Total runtime is given in hours.

Comment 19. Fig 6 shows that all methods still do quite badly: it would be useful to analyze what the main reasons are.

We fully agree that none of the tested tools, including EDTA, RepeatModeler2, or their combination with TEtrimmer, were able to recover the manually curated reference TE libraries fully, despite notable improvements introduced by TEtrimmer. This reflects the inherent challenges of *de novo* TE annotation. Notably, our benchmarking results align well with previously published evaluations (Flynn et al., 2020; Hu et al., 2024 DOI: 10.1038/s41467-024-49912-8; Rodriguez & Makałowski, 2022), further supporting the validity of our benchmarking approach. As also outlined in the Introduction of our manuscript (lines 95-132), there are several underlying reasons for this limited performance:

(1) Structure-based methods, such as EDTA, are highly effective for detecting TEs with well-defined structural features, such as LTR retrotransposons or DNA transposons with terminal inverted repeats (TIRs). However, EDTA is less effective for elements like LINES and SINES, which lack distinct structural motifs (Gozashti & Hoekstra, 2024; Ou et al., 2019, 2024).

(2) Repeat-based methods like RepeatModeler2, RepeatScout, and RECON rely on the presence of multiple similar copies to generate consensus sequences. Consequently, these methods tend to underperform on low-copy-number TEs or highly-divergent TEs. TETrimmer also relies on the repetitive nature of TE sequences to construct high-quality consensus sequences. When TE copies are too few or too degraded, accurate reconstruction becomes challenging. By contrast, the manually curated TE libraries used as the reference in our benchmarking approach are the result of years of expert-guided curation, involving in-depth structural and sequence analysis, and are able to identify nested, degenerated, and low-copy-number TEs that often escape automated detection. To address these limitations, we designed TETrimmer to automate the manual curation of TEs. While TETrimmer significantly enhances the recovery of high-quality TE sequences, a gap still remains. To further bridge this gap, we introduced a user-friendly graphical interface tool (see **Figure 5** in the manuscript) that enables rapid inspection and refinement of TETrimmer outputs. This allows users to approach quickly the level of manual curation of the TE consensus library. We nonetheless added a brief discussion of the inherent challenges of *de novo* TE annotation to our manuscript (lines 693-702).

Reviewer #1 (Remarks on code availability):

I have briefly looked at the code website, and it does seem to be available.

Reviewer #2 (Remarks to the Author):

The manuscript titled “TEtrimmer: a novel tool to automate the manual curation of transposable elements” by Qian et al. describes a novel tool to automate some steps of the laborious manual curation process to lower the barrier of entry to high-quality TE curation and aid in improving the generation of gold-standard TE resources. The manuscript is well-written and clearly outlines the rationale, approach, and findings. This is a valuable contribution to the TE community, and I commend the authors for this effort.

I have a few comments and questions that I would like the authors to clarify before acceptance. Importantly, several claims throughout the manuscript require statistical comparisons for support, as they are currently lacking in some sections. Additionally, increased transparency in specific areas would help researchers assess the tool’s usefulness for their needs.

We thank the reviewer for their encouraging feedback.

Please note that we numbered the comments in order to better refer to them.

Main Comments

Comment 1. - On line 185, it states that “These criteria might not efficiently evaluate TEs lacking terminal repeats such as LINEs and SINEs”. Does this mean that a LINE or SINE will never be evaluated by TEtrimmer as “perfect” or “good”? Why was the decision taken to evaluate outputs in this manner? Would it be possible to implement a switch so that evaluation accounts for TE classification? For example, if a consensus is classified as a LINE/SINE, could the tool prioritize checking for 3’ tail characteristics instead of terminal repeats? Given the quality of the tool and the developers involved, I imagine this would be feasible to implement—though it could also be a future improvement rather than a requirement for initial publication. This addition would be particularly beneficial for non-specialists.

Yes, LINEs and SINEs will not be classified as “Perfect” and “Good” by TEtrimmer. The 5’ end of LINEs is frequently truncated (Goubert et al., 2022; Hartig et al., 2023 DOI: 10.1111/tpj.16208), so LINEs typically have low numbers of intact copies in the genome. In this case, TEtrimmer may not define the correct 5’ end boundaries of LINEs (see also our response to reviewer 2 Comment 6 below). As for SINEs, there is also no conserved sequence feature that can be used to define the accurate 5’ end boundary. The definition of “Perfect” or “Good” in the TEtrimmer evaluation system implies that outputs are comparable to the traditional manual curation of TEs, and it is not necessary to perform manual inspection for these consensus sequences with the provided GUI tool. Since the 5’ end boundary definition of LINEs and SINEs is not possible computationally, we always place them in the “Recommend_check” and “Need_check” categories, so as to not mislead the users and encourage them to inspect LINEs and SINEs manually with the provided GUI tool.

For the 3’ end boundary definition, TEtrimmer examines if the term “LINE” or “SINE” exists in the TE classification name (lines 889-891) and performs poly-A pattern detection. The previous version of TEtrimmer only searched for the presence of the 3’ poly-A tail of LINEs and SINEs. Inspired by the minor comment (reviewer 2 Comment 10), we have upgraded this function and added two more parameters, including “--poly_pattern” and “--poly_len”. This improvement now enables TEtrimmer to identify user-defined multiple 3’ poly-patterns, like poly-A, poly-T, or minisatellite tails of LINEs and SINEs. For example, users can provide multiple poly-base patterns in a comma-separated list, like: A,T,TAAT, where the order of patterns determines the priority for the search. In this example, TEtrimmer first will search for the

presence of a poly-A pattern. If a poly-A is not detected, it instead searches for a poly-T tail. If poly-T is also missing, TETrimmer finally examines the sequence for a poly-TAAT tail. The “--poly_len” parameter defines the required minimum length of any poly tail. The default number is set to 10, which means it requires at least 10xTAAT to be regarded as a poly-TAAT tail. However, these functions can only define the 3’ end boundaries of LINEs and SINEs. The 5’ ends of LINEs and SINEs remain difficult to detect. While having a structurally similar ORF and 3’end, LINEs can show high divergence and/or truncations in their 5’-UTRs even within a single species (Hartig et al., 2023 DOI: 10.1111/tpj.16208). Further, SINEs can show multimerization of their 5’ ends, which again makes them difficult to detect without manual inspection (Seibt et al., 2020 DOI: 10.1111/tpj.14567). Therefore, we still place them into the categories “Recommend_check” and “Need_check”. We changed the manuscript to highlight these issues (lines 190-192, 451-452, and 660-662). In the future, we will consider improving the LINE and SINE curation process further and add a separate scoring system for Non-LTR elements.

Comment 2. - Table 1: Only relatively high-copy-number TEs (≥ 30 sequences) are evaluated as “perfect.”. However, copy number alone does not necessarily indicate a better consensus sequence. Was this threshold based on observations during testing? I can imagine that recently active TEs might lead to “perfect” evaluations due to the presence of many intact copies. To what extent do the authors observe “perfect” TE consensus with MSAs of < 30 sequences? It would be valuable to discuss how TETrimmer performs in such cases. A brief justification in the Methods section or a short 1–2 sentence explanation of why these cutoffs were chosen would improve transparency.

We fully agree that a higher copy count does not necessarily guarantee a better TE sequence. However, as TETrimmer primarily curates TE sequences by leveraging the multi-copy nature of TEs, a higher sequence number increases the likelihood of obtaining an accurate TE model. In addition to the evaluation level “perfect”, TETrimmer also uses the category “good”, which only requires 10 copies of TE sequences. Both “perfect” and “good” outputs are comparable to traditional manually curated library sequences. We also provided another parameter called “--define_perfect” to allow users to define their minimum TE copy number for a TE consensus sequence to be categorized as “perfect”.

We used our benchmarking data (**Figure 6** in the manuscript) and identified the number of TE sequences that were evaluated as “perfect” by the used benchmarking method (Flynn et al., 2020) (please note the difference between the “perfect” evaluation assigned by TETrimmer, and the “perfect” definition used for benchmarking) but had less than 30 copies. We found that between 12% (*H. sapiens*) and 53% (*B. hordei*) of these TE sequences were “perfect” (accurate) (**Reviewer-only Table 2**). This shows that the TETrimmer-intrinsic qualification of TE consensus sequences based on copy number is valid, as the accuracy level varies heavily for low-copy TEs depending on the input genome.

Reviewer-only Table 2: Accurate (“perfect”) TE consensus sequences generated by less than 30 sequences.

Species	RM2+TETrimmer analyzed TE sequences number corresponding to benchmarking evaluation	RM2+TETrimmer analyzed TE sequences number corresponding to benchmarking evaluation with < 30 copies	Percentage
Blumeria hordei	229	121	53%

Drosophila melanogaster	87	42	48%
Danio rerio	749	218	29%
Oryza sativa	491	245	50%
Zea mays	409	187	46%
Homo sapiens	122	15	12%

Comment 3. o (i) How effective is this approach for older, fragmented, or highly diverged TE families? Would clustering produce multiple noise clusters or fail to form clusters altogether? In scenarios where BLAST-extracted copies are highly diverged but still generate BLAST hits, do the authors have recommendations?

Compared to young TEs, older and highly diverged TEs are more difficult to cluster. However, TETrimmer also introduces a method to consider special scenarios. In general, if the TEs are too old and divergent, the TETrimmer cluster module will classify the majority of the sequences into a “Noise (cluster -1)” cluster (see new **Supplementary Figure 7D**). If the sequence number in the noise cluster is higher than 15 and occupies more than 60% of the total sequence of the MSA, and if the “normal” cluster number is smaller than `--max_cluster_num`, the “Noise” cluster will be used for further analysis. For example, TE “`rnd_1_family_420`” is a DNA element from *B. hordei* initially identified by RepeatModeler2. This element is quite divergent, and the TETrimmer clustering module classified 97 sequences into the “noise” cluster (**Supplementary Figure 7D**). In this case, the “Noise” cluster was used for further analysis like MSA extension, MSA cleaning, and TE boundary definition, and the final result can be found in **Supplementary Figure 7A, B, C, and E**. Even though the clustering for older TEs may still fail, the input sequence will be classified as “skipped”, or “low_copy”, and the users can inspect them easily with the provided GUI tool. We changed the manuscript to highlight these clustering strategies for older TEs in lines 239-250 and 798-803) and added this scenario to a new Supplementary Figure (**Supplementary Figure 7**).

Comment 4. o (ii) [Fig. 2 and lines 239-240] Does the clustering process incorporate the original consensus sequence during the initial alignment, or does it rely solely on genomic copies? I have encountered cases where the final consensus shifts toward an abundant repetitive sequence adjacent to the original consensus in repeat-rich genomes. Does TETrimmer account for this possibility, particularly in the cleaning process mentioned in Line 242? If not, could such biases affect the final output?

We appreciate pointing out this limitation. TETrimmer does not include the original query TE consensus sequence during the initial alignment. Because TETrimmer extends both ends of each BLASTN hit to identify the potential boundaries of TE copies, a shift in the curated consensus sequence may occur when the input TE sequence is adjacent to a high-copy TE, and if this specific arrangement exists in more than 10 genomic instances. In such rare cases, the high-copy TE region may dominate the alignment and skew the resulting consensus sequence. We have observed such an occurrence only once during the testing (in the case of a solo LTR). Nonetheless, this is admittedly a limitation of TETrimmer and possibly other tools (e.g. MCHelper). Fortunately, such cases can be easily identified through the report dot plots (see e.g., **Figure 4F** in the manuscript). We recommend that users inspect the corresponding query TE manually using the provided GUI tool for this limitation.

Comment 5. o (iii) The default setting resolves two clusters (Lines 666-667), and the authors state that increasing this number can lead to false positives. However, Figure 2 depicts 13 clusters plus a noise cluster, which may mislead readers into assuming this is typical behavior. While I understand the example was chosen to showcase the tool’s potential, it would be beneficial to clarify in the figure legend or text that the default behavior is not to resolve all possible clusters

(and the reasons why). Additionally, are all 13 clusters in Figure 2 true TE subfamilies? How does TETrimmer define these subfamilies?

We addressed the question regarding the default number of clusters in the response to comment 18 of reviewer 1 (see above).

The TE family term is defined using the 80-80-80 rule (Wicker et al., 2007). A commonly accepted understanding of TE sub-families is that TE sequences classified into the same family can still be further grouped into distinct sub-groups based on higher sequence similarity. However, the precise definition of TE sub-families remains under discussion. To avoid confusion, we deleted all mentions of “sub-family” from the manuscript and replaced them with alternative phrases (lines 141-143, 210-211, 214, 217, 229 and 775-778).

Comment 6. - Lines 250-251 and 703-704. We often find LINEs where 1 or 2 copies are intact, with the majority being truncated. How does TETrimmer handle these cases? Here, the gap percentage could be >90%. Does this lead to truncation of full-length LINEs? If not, what safeguards are in place to prevent this? These scenarios are important to highlight, and the authors could take the opportunity to provide their recommendations (in the methods, or perhaps in the GitHub repository).

When a LINE has only 1 or 2 intact copies, TETrimmer cannot identify the correct 5' end boundary of LINEs (see also our reply to reviewer 2 Comment 1). By default, TETrimmer eliminates the columns that have less than five nucleotides, potentially leading to truncated 5' ends of LINE and SINE elements. As defined by the evaluation system of TETrimmer (**Supplementary Table 1**), LINE elements will not be classified as “Perfect” or “Good” even though we introduced an improved LINE 3' end poly-pattern identification function, which can help to find the right 3' end boundary of LINEs (see also our reply to reviewer 2 Comment 1). As a result, LINE elements are typically classified as “Recommend_check” or “Need_check”. Hence, we strongly advise users to inspect and improve manually the LINE and SINE annotations with the provided GUI tool. We added a respective recommendation to the manuscript (lines 190-193, 451-452, and 660-662), and in the TETrimmer GitHub repository (README.md).

Comment 7. - Lines 257-259: The manuscript presents a single example of a high-copy-number LINE, but this does not represent the typical TE curation challenges we often face. The authors mention that `crop_end_by_gap` is better suited for LINEs, but how does its performance compare to `crop_end_by_divergence`? Are both equally effective for LINEs? Could TETrimmer use them dynamically without requiring the user to choose? Additionally, it would be helpful to include an example of a less ideal case (e.g., a LINE with few intact copies and many fragments) to illustrate tool performance in more realistic scenarios.

We agree that LINE elements with low copy numbers are challenging to annotate correctly. Compared to other types of TEs, the 5' end of LINE sequences are frequently truncated (Hartig et al., 2023 DOI: 10.1111/tpj.16208), which potentially leads to more lowly conserved alignment regions (**Supplementary Figure 1A**). To clean the LINE-based MSA, TETrimmer automatically applies both “`crop_end_by_gap`” and “`crop_end_by_divergence`” functions when the keyword “LINE” is identified in the TE classification name without the requirement that users have to choose it (lines 889-891). For other types of TEs, only “`crop_end_by_divergence`” is used to clean the corresponding MSA, which is sufficient. We demonstrate this capacity exemplarily for a LINE with few intact copies and many fragmented sequences. The example LINE is named “`rnd_5_family_5398`”, which was originally identified by RepeatModeler2 in the genome of *D. rerio*. In the example, 18 sequences are included in the MSA. The example shows the combination of “`crop_end_by_gap`” and “`crop_end_by_divergence`”, which efficiently cleaned this LINE element with fewer intact copies and more fragments than the other

example shown in the manuscript. We provide this example as new **Supplementary Figure 1** and discuss this case in the manuscript (lines 239-250).

Comment 8. - Lines 330-331: The manuscript states that TETrimmer removed 4,835 bp from the consensus, eliminating one ORF and fusing two others. Maintaining intact ORFs is crucial for TE classification, so this raises concerns. In Figure 4E, the gag ORF appears to be missing, yet intact LTR retrotransposons should have both gag and pol. Is this fusion justified? Does the final consensus match expected biological structures? This issue requires further validation and justification what the TETrimmer output is expected.

We fully agree that an intact, autonomous LTR retrotransposon must contain both gag and pol protein domains, which are required for its transposition. In our example element shown in manuscript **Figure 4E**, the input TE consensus sequence “ltr-1-family-22” is a nested TE. After TETrimmer analysis with `--max_msa_line = 500` (default 100), two different TE consensus sequences were generated. One output, named as “ltr-1-family-22-output-1” a LTR retrotransposon, was derived from the internal input sequence (from 1,167 bp to 6,345 bp; see also **Reviewer-only Figure 1**), the other output, named as “ltr-1-family-22-output-2” a LTR retrotransposon, was derived from the remaining region of the input TE sequence (**Figure 4** in the manuscript).

After separation of “ltr-1-family-22-output-1” and “ltr-1-family-22-output-2”, the associated protein domains were also separated. For example, the protein domains from 1,167 bp to 6,345 bp of the input TE sequences (see also **Figure 4E** upper panel and **Reviewer-only Figure 1E** upper panel before TETrimmer analysis) were assigned to “ltr-1-family-22-output-1”.

As for “ltr-1-family-22-output-2”, the gag protein domain was not identified initially (old **Figure 4E** in the manuscript and **Reviewer-only Figure 2A**). The reason for this is that we initially used the default and relatively stringent e-value parameter (e^{-5}) for `pfam_scan.pl`, which was employed by TETrimmer for indexing the PFAM database. By contrast, we identified the gag protein domain in the TETrimmer-curated sequence “ltr-1-family-22-output-2” when we decreased the stringency of the respective PFAM domain search e-value to $1e^{-2}$ (see **Reviewer-only Figure 2B**). Therefore, we have now modified the TETrimmer code by changing this default parameter to $1e^{-2}$, a threshold that was also adopted by Nakamoto and co-workers (Nakamoto et al., 2023 DOI: 10.1093/gbe/evad206) for the TE protein domain prediction. This is also highlighted in line 908 of the revised manuscript. Hence, the initial absence of the gag protein domain is **not** due to its non-existence, but because of unsuccessful prediction. We also believe that even with a more permissive e-value threshold in `pfam_scan.pl`, not all TE proteins can be detected due to limitations of the PFAM database. Therefore, at least for now, we cannot heavily rely on the completeness of PFAM-based TE protein domain predictions to assess TE quality. For future updates, we consider implementing the REXdb, a more comprehensive and refined domain database (Neumann et al., 2019 DOI: 10.1186/s13100-018-0144-1), alongside the PFAM database for protein domain prediction.

A**B****C****D****E****F**
Reviewer-only Figure 1. The input TE “ltr-1-family-22” is a nested TE. After TETrimmer analysis “ltr-1-family-22-output-1” was generated derived on the internal input TE sequence. The *B. hordei* LTR retrotransposon named “ltr-1-family-22”, initially annotated by RepeatModeler2 and re-analyzed with TETrimmer. **A** The alignment of 44 sequences in total shows the TE boundary regions of the MSA (100 nucleotide sites are displayed for each side) after TETrimmer analysis. Nucleotide background colors in the MSA represent sites where the proportion of the respective nucleotide is below 0.4. The TE boundaries defined by TETrimmer are indicated as “TE start” (red) and “TE end” (blue). The two boundary regions were artificially connected by ten gaps (“-”) and the total length of the MSA is indicated on the top. **B** The plot shows the entire MSA after TETrimmer analysis. The TE boundaries are indicated as in (A). Nucleotides are represented with colored bars; gaps are indicated as blank regions in the plot. The x-axis gives the nucleotide position (in bp) within the MSA. **C** The panels show a BLASTN plot before (left panel) and after (right panel) TETrimmer analysis, each following a BLASTN search of the respective TE consensus sequence to identify TE sequence hits at a genome-wide scale. The x-axis indicates the TE consensus nucleotide position (in bp) and the y-axis is the sequence divergence in percent compared to the TE consensus sequence. Each line indicates a BLASTN hit; red lines highlight hits with a sequence divergence below 1.5% and a sequence coverage > 90%. **D** Self-dot plots before (left panel) and after (right panel) TETrimmer analysis. The axes show the nucleotide position (in bp) in the TE consensus sequence. The repeat regions are represented by short diagonal lines outside the main diagonal. **E** The plot shows a genomic map of open reading frames (ORFs; blue arrows) and PFAM domains (red arrows) predicted in the TE consensus sequence before (upper panel) and after (lower panel) TETrimmer analysis. The direction of the arrows indicates the strands of the respective ORFs/PFAMs in the sequence. ORF and PFAM identity are indicated on top of the arrows. **F** Dot plots (window size = 25 bp, threshold = 50; the threshold represents the minimal sum of substitution scores in a defined window required for a dot to be plotted) of the TE consensus sequence before (y-axis) and after (x-axis) TETrimmer analysis. Axes show the nucleotide position (in bp). Identical regions are represented by short diagonal lines outside the main diagonal.

Reviewer-only Figure 2. The gag protein domain of “ltr-1-family-22-output-2” was identified when the e-value of pfam_scan.pl was decreased. The plots show a genomic map of open reading frames (ORFs; blue arrows) and protein family (PFAM) domains (red arrows) predicted in the TE consensus sequence before (upper panel) and after (lower panel) TETrimmer analysis. The direction of the arrows indicates the strands of the respective ORFs/PFAMs in the sequence. ORF and PFAM identities are indicated on top of the arrows. **A** An e value 1e-5 was used for pfam_scan.pl search. The upper panel represent the input TE consensus sequence. The lower panel was one of the outputs after TETrimmer analysis “ltr-1-family-22-output-2”. **B** An e value 1e-2 was used for pfam_scan.pl search. The upper panel represent the input TE consensus sequence. The lower panel was one of the outputs after TETrimmer analysis “ltr-1-family-22-output-2”.

Comment 9. - Lines 434, 443, 490: The manuscript makes qualitative claims (e.g., “considerably”) without statistical support. These should be backed by appropriate statistical tests, which could be reported in the text and/or visualized in Figure 6. For example, are increases in perfect/good/present/not found TE families statistically significant? Simple statistical tests would improve transparency and add support to the findings.

- Figure 7: The claim that TEtrimmer provides “superior genome-wide TE annotation ability” requires statistical backing. Figure 7B indicates classification conflicts between tools and reference annotations. Has misclassification been significantly reduced? Statistical comparisons of annotation improvements (e.g., total annotated base pairs, classification accuracy) should be provided.

We agree that statistical support can be useful to substantiate such hypotheses/statements. However, in the absence of independent replication, which, due to the computational resource/time requirements for some of the genomes (esp. human and maize) is not possible to obtain within the time limit granted for these revisions, we cannot provide a reliable statistical significance analysis in these cases. Therefore, we refrain from using phrases indicating statistical support (considerably, significantly, superior ...) and removed these instances from the document (lines 487, 537, 554, 564, 621, and 648).

Minor Comments

Comment 10. - Line 67: The manuscript states, “LINEs and SINEs are defined by their genomic repeats.” Is this always the case? The Wicker et al. (2007) classification mentions additional characteristics (e.g., AT-rich tails, tandem repeats, poly(T) tails). In my experience, tandem repeats are common in some taxa. Clarifying this would strengthen the manuscript. As a side note, does this have any impact on the structure-based modules of TEtrimmer? Does it only look for poly-A signals?

The manuscript text has been updated to “LINEs and SINEs are defined by their terminal genomic poly-A, poly-T, or microsatellite tails” (lines 75-76). A detailed description regarding the improvements of TEtrimmer to detect SINEs and LINEs is provided above (reviewer 2 Comment 1).

Comment 11. - Line 287: What are the cutoffs that define lowly conserved regions? These could be defined in the text.

In general, the lowly conserved regions represent the poorly aligned regions in the MSA (lines 995-996). TEtrimmer defines the cutoffs of lowly conserved MSA regions by the MSA cleaning functions, including “crop_end_by_divergence” and “crop_end_by_gap”. All MSA regions that can be cleaned by these two functions with the default parameters are defined as lowly conserved regions. The default parameter are

```
--crop_end_div_thr = 0.7, --crop_end_div_win = 40,  
--crop_end_gap_thr = 0.1, and --crop_end_gap_win = 250.
```

We illustrated these two functions in the manuscript (lines 805-840). A detailed description of the two functions can be found in the provided tutorial video named “TEtrimmer background introduction“ from time 18:06 to 23:15. Links:

<https://www.youtube.com/watch?v=PgKw76gmzI8>

or

https://www.bilibili.com/video/BV1Y5NDePEq/?share_source=copy_web&vd_source=e97586c562998df25f9322fd7a2705e6

Comment 12. - Lines 661-662: How are indels represented in the pseudo-sequences representing divergent regions? If my understanding is correct, gaps are not counted towards the divergence calculation. In this case, is it possible that an indel column could have a nucleotide proportion >0.8 but represent a true subfamily (as the indel is present in a subset of sequences and

might have high identity among those sequences containing the indel). Does this mean this column containing an indel position will not be counted as a variable region? How would this impact the inference of subfamilies? Often in manual curation, these are the easier subfamilies to define, so it is important that indels are features that can help to define subfamilies.

We thank the reviewer for highlighting the importance of indels in defining TE subfamilies and MSA clustering. During the MSA clustering process of TETrimmer, the MSA columns containing an indel position are indeed treated as variable regions, which are named as “gap blocks” (see new **Supplementary Figure 7**. To identify and handle gap blocks, TETrimmer uses the function “select_gaps_block_with_similarity_check” (lines 780-786). A gap block is defined as a contiguous set of MSA columns that satisfied both of the following criteria:

1. Gap proportion consistency: The difference in gap proportions between adjacent columns within the block does not exceed 10%.
2. Boundary nucleotide conservation: The proportion of the dominant nucleotide at each boundary of the gap block must be >0.8 , ensuring the block is flanked by conserved regions (for an example, see new **Supplementary Figure 6**).

Thereafter, the MSA columns from the gap block are combined with the divergent MSA columns to create a pseudo-MSA, which is subsequently used for MSA clustering. Initially, we did not mention this in the manuscript, but we have now added a description to clarify this point in the revised version (lines 780-786).

Reviewer #2 (Remarks on code availability):

The code is well-commented and easy to follow. Descriptions are provided where appropriate, and data is reproducible. The README is well-structured and provides enough guidance for usage of the tool.

We thank the reviewer for the positive assessment of our documentation.

Reviewer #3 (Remarks to the Author):

Recommendation: Revision

Summary:

This manuscript introduces TEtrimmer, a novel tool designed to automate manual transposable element (TE) curation. TEtrimmer addresses key limitations in existing tools by enhancing multiple sequence alignment (MSA) clustering, sequence cleaning, and TE boundary definition. The tool features a graphical user interface (GUI) and report generation, improving both usability and annotation accuracy. The authors provide a quantitative comparison of TEtrimmer's performance against existing methods, demonstrating that TEtrimmer enhances TE annotation precision while reducing manual workload. Overall, this manuscript presents a comprehensive and well-structured study on TE annotation and curation, effectively justifying the need for TEtrimmer. I also explored the GitHub repository and the YouTube instruction video, and I appreciate the detailed documentation, which will be highly valuable to future users. Below, I outline several questions and comments that could help further improve the manuscript:

We are grateful for the positive and constructive feedback by the reviewer.

1. Justification for MSA-Based TE Classification. MSA sequence-to-sequence alignment based classification can be computationally expensive. Why is MSA preferred over other sequence clustering methods, such as Hidden Markov Models (HMMs) or machine learning-based approaches? What are the benefits to use MSA as major algorithm in TEtrimmer?

We appreciate that the computation cost of multiple sequence alignment (MSA)-based clustering is a very relevant point for users. Below, we clarify why MSA is preferred over alternative approaches. HMMs rely on pre-aligned MSAs for model/matrix construction. Thus, they do not bypass the computational cost of MSA. To evaluate the clustering performance, we also compared the TEtrimmer MSA clustering module with the first stand-alone HMM-based DNA sequence clustering tool, `hmmcluster` (Charles et al., 2023 DOI: 10.1073/pnas.2221797120), on the same example dataset used in the manuscript (**Figure 2**). As a result, `hmmcluster` only provided three clusters, which were insufficient to separate TE subfamilies properly. We also tried alignment-free software like CD-HIT-EST, which uses a k-mer-based greedy algorithm (Fu et al., 2012). However, like the HMM-based approach, this method does not properly cluster TE-related MSAs and does not separate TE variants. In fact, the TEtrimmer MSA clustering module employs the machine-learning-based method DBSCAN (density-based spatial clustering of applications with noise) (Birant & Kut, 2007), which can automatically determine the optimal cluster number and remove noise sequences from the MSA.

MSA is not only fundamental for clustering but also essential for the cleaning and TE boundary definition modules. TEtrimmer introduces a sliding window-based strategy for cleaning of the lowly conserved regions of TE sequences and includes the functions `crop_end_by_divergence` and `crop_end_by_gap`. Both functions rely on an MSA. For example, the `crop_end_by_divergence` function needs to convert the MSA to a proportion matrix, which is not feasible with HMMs. To identify the intact TE sequences, TEtrimmer iteratively extends both ends of sequences using the MSA and determines the extension size, which cannot be done based on HMMs. Furthermore, the use of MSA enables user inspection and manual refinement of TEtrimmer results *via* the provided GUI tool. HMM models, by contrast, are not easily inspectable or modifiable. We acknowledge that HMMs have superior performance for storing and indexing sequence alignment databases, which is why they are used in databases such as Dfam and PFAM (Hubley et al., 2016; Mistry et al., 2021). To leverage this strength,

TEtrimmer includes the “--hmm” parameter, allowing users to generate HMM models from the final TEtrimmer-curated MSA output.

2. Could CD-HIT-EST miss TE subfamilies during de-duplication, particularly if the clustering threshold is too high or if subfamilies are highly similar? Some TE subfamilies diverge due to mutations and indels, and CD-HIT-EST may overlook rare or ancestral subfamilies. Additionally, fragmented TEs might be erroneously clustered into separate groups instead of being assigned to their correct subfamily. Can the authors comment on this issue?

If this comment refers to the de-duplication of the input file (lines 757-758) and is related to the parameter “--dedup”: TEtrimmer uses CD-HIT-EST for the de-duplication of the input TE consensus library when the “--dedup” parameter is specified. We use stringent criteria for the de-duplication of the input TE consensus library. This process merges sequences that have over 90% identity and more than 95% coverage. For example, in **Reviewer-only Figure 3**, TE sequences 1, 2, and 3 are from the input TE consensus library (--input_file) and can be aligned. After de-duplication, only TE sequence 2 is eliminated.

Reviewer-only Figure 3. Schematic of TEtrimmer’s de-duplication strategy for an input TE library. Each TE sequence is shown as a horizontal black line. Aligned regions among TE sequences are marked by small vertical bars.

If this comment is related to the de-duplication of the output TE consensus library after TEtrimmer analysis (lines 939-952): To avoid over-de-duplication and overlook rare or ancestral TEs, TEtrimmer employs two rounds of de-duplication for the output TE consensus sequence.

For the first round of output de-duplication, TE consensus output sequences are grouped if they share more than 90% (-c 0.9) identity and the aligned region occupies over 90% (-aL 0, -aS 0.9) for the shorter sequence. For instance, in **Reviewer-only Figure 4**, TE sequence 1 to 3 are placed in “Group” 1 after CD-HIT-EST round 1, and TE sequences 4 to 7 are gathered in “Group 2” after CD-HIT-EST round 1.

Group 1 after CD-HIT-EST round 1

TE sequence 2 is evaluated as "Perfect". Sequence 2 is selected to represent the Group 1 after CD-HIT-EST round 1. TE sequence 1 and 3 are eliminated after output sequence de-duplication.

Group 2 after CD-HIT-EST round 1

No sequences from the Group 2 after CD-HIT-EST round 1 is classified as "Perfect" or "Good". All sequences from this group are used for the next round de-duplication.

Reviewer-only Figure 4. Schematic of TETrimmer's round 1 de-duplication strategy for an output TE library. Each TE sequence is shown as a horizontal black line. Aligned regions among TE sequences are marked by small vertical bars.

Subsequently, TETrimmer selects groups that contain at least one "Perfect" or "Good" consensus sequence (**Supplementary Table 1**) (Group 1 from **Reviewer-only Figure 4**). For the selected group, the longest "Perfect" consensus sequence (TE sequence 2) is chosen as the representative if "Perfect" candidates exist. Otherwise, the longest "Good" consensus sequence is selected. In the second round of output de-duplication, all sequences from the remaining groups of round 1, like Group 2 in **Reviewer-only Figure 4**, are extracted and reprocessed using CD-HIT-EST. Sequences are clustered again if they have more than 85% (-c 0.85) identity, and the aligned regions occupy more than 80% (-aL 0.8, -aS 0.8) of all sequences in each group (**Reviewer-only Figure 5**). Only the longest sequence in each new cluster is selected. Finally, the selected consensus sequences from two rounds of de-duplication are combined to form the final de-duplicated TE consensus library, which is saved in the file "TETrimmer_consensus_merged.fasta".

Group 1 after CD-HIT-EST round 2

Because TE sequence 4 is longer, TE sequence 5 is eliminated after output sequence de-duplication.

Group 2 after CD-HIT-EST round 2

Because TE sequence 6 is longer, TE sequence 7 is eliminated after output sequence de-duplication.

Reviewer-only Figure 5. Schematic view of the round 2 de-duplication strategy of TETrimmer for the generation of an output TE library. Each TE sequence is shown as a horizontal black line. Small vertical bars mark aligned regions among TE sequences. After the two rounds, sequences 2, 4, and 6 were kept.

All deleted output TE consensus sequences can still be found and inspected by the provided GUI tool. Users can easily rescue the output sequences they want to keep, which provides more flexibility to the users. TETrimmer also provides the TE consensus library before de-duplication named "TETrimmer_consensus.fasta", users can also do their own de-duplication.

3. LINE 157-160: Is it necessary to separate each sequence into an individual file for multi-threading? Creating a large number of small files may introduce I/O bottlenecks. The authors can consider using batch processing instead of file-based separation. It might be a better idea to use multi-threading in a Single BLASTN Job with the built-in multi-thread parameters; This allows BLASTN to internally distribute sequence queries across multiple threads without excessive file I/O overhead.

We concur that the proposed modification would likely enhance both performance and resource usage. Importantly, this change would not affect the biological validity of the TETrimmer results. While it is not implemented in the current version, we will carefully consider incorporating it in future updates. We remain committed to actively improving TETrimmer based on user feedback.

4. Did the authors benchmark computational resources (runtime, memory usage) for TETrimmer against other tools? Given that the TE annotation can be a computationally intensive task, users would benefit from knowing the expected runtime and memory requirements for different genome sizes. At a minimum, the authors could provide this information for model genomes tested in the study.

We agree that the required computational resources are highly relevant information for the users. Unfortunately, we used different CPU cores for each genome and did not record computational resource usage during the initial benchmarking tests. We believe that reporting this information is crucial to help users evaluating hardware requirements and to enable comparisons with similar tools. A comparable tool to TETrimmer is MCHelper (Orozco-Arias et al., 2024). The authors of MCHelper provide runtime tests for four organisms: *D. melanogaster*, *O. sativa*, *D. rerio*, and *C. cornix*. According to the MCHelper publication, their analyses were performed using 48 CPU cores on a compute node with a maximum of 240 GB RAM. To allow a direct comparison with MCHelper, we tested TETrimmer runtime on four genomes: *D. melanogaster*, *D. rerio*, *O. sativa*, and *B. hordei*. Each analysis was executed using 48 CPU cores and 140 GB of RAM. Three of the four genomes (*D. melanogaster*, *D. rerio*, and *O. sativa*) are the same as the ones used for the MCHelper benchmarking, enabling a general performance comparison. Below, we provide the TETrimmer runtime results and output folder sizes for both the RepeatModeler2 (RM2) and EDTA input TE libraries. We also added this information as a new table in the revised manuscript version (new **Table 2**):

Table 2. Runtime test of TETrimmer for the genomes of four organisms.

Species	Genome size (Mbp)	EDTA as input for TETrimmer			RM2 as input for TETrimmer		
		Input TE number	Runtime (h) ¹	Output folder size (GB)	Input TE number	Runtime (h) ¹	Output folder size (GB)
B. hordei	124	996	0.92 ± 0.049	2.30	818	0.83 ± 0.040	2.10
D. melanogaster	144	819	0.66 ± 0.067	0.92	480	0.66 ± 0.046	0.96
D. rerio	1,679	8,631	4.95 ± 0.225	15.10	3,504	2.32 ± 0.066	5.50
O. sativa	373	10,404	3.30 ± 0.200	7.30	2,334	1.31 ± 0.090	2.50

¹ TETrimmer was executed 3 times for each TE library with 48 CPUs and 140 GB RAM. The runtime average and standard deviation are shown.

For comparison, we show here the MCHelper runtime results (**Reviewer-only Table 3**; Orozco-Arias et al., 2024).

Reviewer-only Table 3. MCHelper runtime results, taken from (Orozco-Arias et al., 2024).

Species	RM2		EDTA		REPET	
	Total runtime (h)	Extension step runtime (%)	Total runtime (h)	Extension step runtime (%)	Total runtime (h)	Extension step runtime (%)
D. melanogaster	1.42±0.004	76.87	1.10±0.025	62.02	3.46±0.004	81.67
O. sativa	10.82±0.050	83.64	10.71±0.052	82.28	16.55±0.084	79.76
C. cornix	5.98±1.067	72.34	7.52±0.033	63.48	12.33±0.439	54.85
D. rerio	29.91±0.094	95.91	41.95±0.1056	72.92	27.92±0.60	70.12

MCHelper was executed 10 times for each raw library (RM2 EDTA and REPET) and the average and standard deviation (SD) were calculated. Total runtimes are given in hours.

Despite using nearly identical computational resources, TETrimmer requires a much shorter runtime than MCHelper for the three shared genomes, *D. melanogaster*, *D. rerio*, and *O. sativa*, based on the input of RM2 or EDTA. For instance, MCHelper required nearly 30 hours to process the *D. rerio* genome using the RM2 library, whereas TETrimmer completed the same task in just 2.3 hours. Notably, Orozco-Arias et al. reported that up to 95.91% of its runtime for *D. rerio* was consumed during the MSA extension step. Based on this information, we assume that the major runtime difference can be attributed to several optimizations in TETrimmer's MSA extension strategy (described in detail in the Methods section "MSA sequence extension" lines 841-879). The TETrimmer MSA extension strategy is also described in the provided tutorial video,

1) "Introduction to TETrimmer Parameters" from time 20:42 to 24:05 min.

Links: <https://youtu.be/8jp3j5FFf1w>

or

https://www.bilibili.com/video/BV18c59zpEZZ/?share_source=copy_web&vd_source=e97586c562998df25f9322fd7a2705e6

2) "TETrimmer Background Introduction" from time 10:14 to 15:07.

Links: <https://www.youtube.com/watch?v=PgKw76gmz18>

or

https://www.bilibili.com/video/BV1Y5NDDePEq/?share_source=copy_web&vd_source=e97586c562998df25f9322fd7a2705e6

5. LINE 416-429 and table 2. The authors used different TE reference libraries for each genome. While I understand that a custom TE library may be necessary for *Blumeria hordei*, why was RepBase not used for the human genome? This could make the comparison more reliable. It is also mentioned in line 486-487 "Due to the overall poor performance on TE consensus library construction for *H. sapiens* by all tested tools (Fig. 6A) and TETrimmer's relatively lower TE annotation score for *H. sapiens*" Would it be possible to improve the performance for human genome by using the RepBase human TE library?

We agree that using Repbase for generating the *H. sapiens* TE library could make the comparison more reliable. However, we preferred using the curated Dfam *H. sapiens* TE library because we think it provides a more comprehensive reference library than the Repbase *H. sapiens* TE database. Before benchmarking the six eucaryotic organisms used in our manuscript, we conducted a thorough search and evaluation of available TE reference libraries, including Dfam 3.7 (curated), Repbase 28.10, and other resources (**Reviewer-only Table 4**).

Reviewer-only Table 4. Comparison of TE consensus libraries.

Species	Dfam 3.7 curated		Rebase 28.10		Other source	
	TE number	Library size	TE number	Library size	TE number	Library size
B. hordei	Not identified	Not identified	257	762,710 bp	465	1,495,021 bp¹
D. melanogaster	222	641,833 bp	296	883,820 bp		
D. rerio	1,986	5,158,108 bp	2,473	6,946,185 bp		
O. sativa	Not identified	Not identified	3,439	8,295,508 bp	2,431	5,101,558 bp ²
Z. mays	Not identified	Not identified	2,106	6,778,752 bp	1,995	5,101,003 bp ³
H. sapiens	1,403	1,690,780 bp⁴	538	846,191 bp		

¹ In-house manually curated TE consensus library.

² Downloaded from <https://github.com/oushujun/riceTElib>

³ Downloaded from <https://github.com/oushujun/MTEC>

⁴ Libraries in bold were selected as reference TE library for the TETrimmer benchmarking analysis.

Our comparison revealed that the Rebase *H. sapiens* TE library contains only 538 consensus sequences with a cumulative size of 846,191 bp. By contrast, the Dfam 3.7 (curated) *H. sapiens* library contains 1,403 consensus sequences with a total length of 1,690,780 bp. Dfam offers greater coverage and sequence diversity for *H. sapiens* TE database. For completeness, we also performed benchmarking using the Rebase *H. sapiens* TE library (**Reviewer-only Figure 6**). We observed the same overall trend: most *H. sapiens* TE consensus sequences were still not recovered. Moreover, the relative performance of the tested tools remained consistent between the two libraries.

Reviewer-only Figure 6. Benchmarking the performance of TETrimmer on Dfam- and Rebase-derived *H. sapiens* TE libraries. We benchmarked the performance of EDTA, RepeatModeler2 (RM2), and both these tools after additional TETrimmer analysis (EDTA+TETrimmer and RM2+TETrimmer, respectively) by comparing the TE quality in the *de novo*-generated consensus libraries with the quality in the corresponding *H. sapiens* reference TE consensus libraries including Dfam 3.7 (curated) *H. sapiens* (left panel), and Rebase 20.10 *H. sapiens* (right panel). The stacked bar graphs show the proportion of correctly discovered consensus TEs by the respective analysis tools/pipelines indicated on the x-axis according to the colour code shown on the right. Benchmarking categories included “Perfect” (pink), “Good” (green), “Present” (orange), and “Not found” (grey) (Flynn et al., 2020). “Perfect” means that the reference TE consensus sequences have one exact match in the *de novo*-generated library with > 95% similarity and coverage. “Good” indicates that the reference TE consensus sequence has multiple overlapping matches in the *de novo*-generated library, each with > 95% similarity and full coverage. “Present” is similar to “Good”, but the required minimum similarity and total coverage is decreased to 80%. The remaining TE sequences were assigned to the “Not found” category.

6. Did the authors evaluate TETrimmer’s performance on solo LTRs and nested TEs? These elements are common in large genomes and they are challenging for annotation. If tested, could the authors provide a discussion on its effectiveness in these more complicated scenarios?

Thank you for this thoughtful question. Solo LTRs and nested TEs indeed pose special challenges in library curation and genome annotation. We have divided our answer into two parts – first, how TETrimmer handles these elements when they appear in the input library, and second, how (or whether) they appear in its output.

1) Treatment in the input library – Because TETrimmer refines an existing repeat library, every input sequence, whether a full-length element, a solo LTR, or a nested TE, is subject to the same multiple-sequence alignment (MSA) clustering and extension steps: Solo LTRs can seed the clustering of their parent LTR retrotransposons. In such cases, the MSA step may “pull in” flanking internal sequences, resulting in a consensus that spans the entire retrotransposon. This however is dependent on genomic organisation of each individual LTR family. Nested TEs in the library likewise enter the MSA clustering. For example, the nested LTR shown in **Figure 4E** (upper panel) served as the seed for one cluster, while the nested element seeded another. Both elements then produced separate consensus in the final output (also see the response to Reviewer 2 Comment 8). The “clean” nesting element can be seen in **Figure 4E** (lower panel) and **Reviewer-only Figure 1E** (lower panel).

2.) Reporting in the output library – After curation (MSA cleaning and extension), TETrimmer tends to produce complete TE consensus sequences rather than partial fragments: Solo LTRs rarely survive as singleton LTRs. Most solo LTRs are either extended into full-length retrotransposons or eliminated if they lack sufficient flanking context or do not show enough copy numbers. Nested elements likewise are typically reported as their full-length consensus; internal fragments alone do not persist as separate outputs. There is no additional reporting on solo LTRs or nested TEs, most copies are consolidated into full-length or primary elements.

To quantify this behaviour, we simulated a genome containing (1) primary TE insertions, (2) true nested insertions, and (3) the corresponding “outer” nesting elements. Using each tool’s library (EDTA, RepeatModeler2, and both followed by TETrimmer), we ran RepeatMasker (-nolow -pa 12 -gff -a -s -inv) on the simulated genome and compared calls to the ground-truth GFF from the simulation process. Concludingly, TETrimmer-curated libraries yield higher recall and precision for nested, nesting, and primary integrations (see **Reviewer-only Figure 7**). This improvement holds across all TE classes present in our simulation with minor exceptions in the TIR superfamily.

Reviewer-only Figure 7: Fraction of correctly covered TE integration sites in a simulated genome using RepeatMasker with *de novo* generated TE libraries with EDTA and RepeatModeler2, followed by curation with TETrimmer. TE integration sites are categorized into (1) true nested, “inner” insertions, (2) the corresponding “outer” nesting insertion site, and (3) primary TE insertions. The annotated genome was a simulated 100 Mb genome containing 75% of TEs. Known TE integration sites (“ground truth”) were compared to RepeatMasker (-nolow -gff -a -s -inv) outputs as previously described in Rodriguez and Makałowski (Rodriguez & Makałowski, 2022). Values plotted above each boxplot contain the mean values ± standard deviation. (RM2, RepeatModeler2; LTR, LTR-retrotransposons (incl. TRIMs and LARDs); DNA, DNA-transposons + MITEs + Helitrons; Other, SINEs + Pararetroviruses).

7. After generating TE consensus libraries, how was whole-genome TE annotation performed? Did the authors apply RepeatMasker for each TE library to conduct genome-wide TE annotation, or was an alternative approach used?

Yes, we used RepeatMasker for the whole-genome TE annotation based on the provided TE consensus library. This is mentioned in the manuscript in lines 1020-1023.

8. It is not clear to me what was the ground truth for benchmarking whole-genome TE annotation? As described in lines 474-485. In the method section, the authors mentioned that all ‘.out’ files were compared with Repeatmasker file. How do the authors justify RepeatMasker annotations as “perfect” ground truth? Would it be feasible to use simulated data for validation? For example, inserting synthetic TE sequences into a TE-free artificial genome and comparing annotation results across tools would provide a controlled accuracy assessment. The simulated TEs insertions are the real ground truth can be used for calculating the matrix.

We fully agree that using RepeatMasker annotations as ground truth harbours certain pitfalls. However, using RepeatMasker for benchmarking *de novo* TE identification is common practice and at least for now the standard, proven in many other publications (Baril et al., 2024; Flynn et al., 2020; Ou et al., 2019). There is a TEhub benchmarking initiative (which we are involved in) trying to resolve these shortcomings in the future.

The RepeatMasker performance is not perfect but known to be congruent to the quality of the input TE library and therefore still provides solid measurements. As input for the “ground truth”, RepeatMasker annotations with manually curated TE libraries have been used (for details see our responses above, using Reprbase, Dfam, etc.). Using manually curated TE libraries has the advantages of reliable input but also implies that many of the results cannot be reproduced by a

fully automated process, as some researchers have decades of experience with certain repeat families. Manual curation is a time-consuming process, which needs a lot of user experience. TEtrimmer aims to assist in this process and allow less experienced users to create similar results. Further, using RepeatMasker with curated libraries means that only known TEs will be taken into consideration and potentially newly discovered TEs will be falsely marked false positives.

We further agree that it is feasible to use simulated genomes as “ground truth” to reduce these kinds of pitfalls. Simulated genomes, however, have their own limitations and can only serve as a model, which is why they were not included previously. Nevertheless, we now include simulated genomes for benchmarking as well (see new panels in **Figure 6** and **Figure 7** in the manuscript) to further highlight the advantages of using TEtrimmer as an additional tool in TE identification and annotation. We simulated three genome sequences with 1) 50 Mb and 50 % repeats, 2) 100 Mb and 50 % repeats and 3) 100 Mb and 75 % repeats. Each of the tools (EDTA, RepeatModeler, and TEtrimmer in addition for each of the first) was run 3 times (runs). The TE libraries were compared to the original TE library (subsample of the BeetRepeatDB (Schmidt et al., 2024)), a manually curated TE library with TEs from all classes of sugar beet and closely related species), which served as input for the simulation. We find that TEtrimmer improves the amount of correctly detected results as well as the completeness of the elements within the resulting TE library. Adding TEtrimmer roughly doubled the compute time in all cases but saves a lot more time compared to a fully manual curation. The TE libraries were used with RepeatMasker and compared to the “ground truth” annotation of the original simulated genome. The resulting genome-wide metrics show similar improvements when adding TEtrimmer to the identification tools by reducing the amount of false negatives and false positives, while improving the true positive and true negative rates (see new **Supplementary Figure 5**).

9. The methods section should include detailed BLASTN parameters to help readers understand the query criteria applied in the first step of TE identification.

We now added the following statement to the methods section of our manuscript: “TEtrimmer uses these separated sequences to perform BLASTN searches against the respective genome to find query copies for each TE sequence with pre-defined parameters, “-evaluate 1e-40, -max_target_seq 10000, -qcov_hsp_perc 15”.” (lines 760-763).

10. Does TEtrimmer perform differently on autonomous vs. non-autonomous TEs? Some steps in TEtrimmer seem tailored for autonomous TEs, which may make them easier to detect. However, non-autonomous TEs are also important to genome evolution. Could the authors clarify whether TEtrimmer is equally effective for both categories?

TEtrimmer includes a step specifically designed for autonomous TEs, which involves PFAM protein domain prediction. The tool uses the orientation of predicted PFAM domains to determine the correct strand direction of the TE (as described in the main manuscript (lines 909-914)). Since non-autonomous TEs typically lack protein-coding domains, this method is not applicable to them. However, TEtrimmer primarily relies on the multi-copy nature of TEs for curation. If the number of intact copies is sufficiently high (≥ 10), TEtrimmer can accurately determine the boundaries of non-autonomous TEs as well.

In conclusion, the manuscript provides a valuable contribution to the field of TE annotation and curation. The authors effectively demonstrate the need for TEtrimmer, and the provided documentation enhances its accessibility. However, addressing the above technical questions, benchmarking details, and methodological justifications would improve the manuscript’s clarity.

Reviewer #3 (Remarks on code availability):

I have reviewed the code, explored the GitHub repository, and gone through the documentation and instructional video. The authors have done a great job organizing the code. They provide a detailed README outlining the download and installation steps.

We thank the reviewer for the positive assessment of our documentation.

REVIEWER COMMENTS

Reviewer #1 (Remarks to the Author):

The authors have addressed my concerns well, and I do not need to re-review this again.

Reply: We welcome the appreciation of our revised manuscript version by this reviewer

I have just one more comment.

I find this part of the Background gratuitously confusing: "Transposable elements (TEs) are selfish repetitive DNA elements that can move... they occupy a large proportion of many genomes, such as around 45% of the human genome".

Non-expert readers (such as my younger self) are misled by such statements into thinking that 45% of the genome consists of intact, active TEs. The plain meaning of "transposable element" is an intact TE.

Reply: We thank the reviewer for pointing out this potentially misleading statement. We have rephrased the sentence to clarify this aspect. It now reads: "Since then, TEs have been identified in almost all studied eukaryotic species where they occupy as either intact or fragmented copies a large proportion of many genomes, such as around 45% of the human genome (International Human Genome Sequencing Consortium et al., 2001), 53% of the zebrafish genome (Howe et al., 2013), and 85% of the maize genome (Schnable et al., 2009)" (lines 57-61 in the revised manuscript version).

We refrain from any statements regarding the extent of fragmented TE copies, as this varies considerably across taxa.

Reviewer #2 (Remarks to the Author):

The authors have addressed my comments with thoughtful and well-formulated responses.

Reply: We welcome the appreciation of our revised manuscript version by this reviewer

I am unable to assess the responses that in particular refer back to new supplementary figures/tables, as the updated ones seem to not have been provided. (e.g. the responses refer to supplementary figure 7 with multiple panels, but supplementary figure 7 in the supplementary

materials provided is a single figure with the title "TEtrimmer utilizes gap information from MSAs to assist in clustering TE sequences.")

Reply: We apologize for this mistake in the previous response letter to the reviewers (related to revision #1). We erroneously referred to the wrong Supplementary Figure in this instance (our reply to comment 3 of reviewer #2). Instead of the wrongly cited Supplementary Figure 7, Supplementary Figure 1 would have been correct here. We double-checked the correctness of all Supplementary Figures in our manuscript and found that this is the only case of such a mixup.

I am supportive of publication of this work, which shows promise in lowering the barrier for manual TE curation, which remains a difficult endeavour.

Reviewer #3 (Remarks to the Author):

The authors have addressed all my comments and questions. The revisions have improved the clarity and quality of the manuscript. I believe the manuscript can now be accepted.

Reply: We welcome the appreciation of our revised manuscript version by this reviewer

Reviewer #3 (Remarks on code availability):

I have reviewed the code and tutorial; the authors provided enough instructions on GitHub.